# Pb-rich Cu grain boundary sites for selective CO-to-n-propanol electroconversion

Wenzhe Niu [1,8], Zheng Chen [2,8], Wen Guo[1], Wei Mao[3], Yi Liu[1], Yunna Guo[4], Jingzhao Chen[4], Rui Huang [1], Lin Kang[1], Yiwen Ma[1], Qisheng Yan[1], Jinyu Ye [5], Chunyu Cui[1], Liqiang Zhang[4], Peng Wang [3,6], Xin Xu [2,7] ✉ & Bo Zhang [1] ✉

Electrochemical carbon monoxide (CO) reduction to high-energy-density fuels provides a potential way for chemical production and intermittent energy storage. As a valuable $C_3$ species, n-propanol still suffers from a relatively low Faradaic efficiency (FE), sluggish conversion rate and poor stability. Herein, we introduce an "atomic size misfit" strategy to modulate active sites, and report a facile synthesis of a Pb-doped Cu catalyst with numerous atomic Pb-concentrated grain boundaries. Operando spectroscopy studies demonstrate that these Pb-rich Cu-grain boundary sites exhibit stable low coordination and can achieve a stronger CO adsorption for a higher surface CO coverage. Using this Pb-Cu catalyst, we achieve a CO-to-n-propanol FE ($FE_{propanol}$) of $47 \pm 3\%$ and a half-cell energy conversion efficiency (EE) of 25% in a flow cell. When applied in a membrane electrode assembly (MEA) device, a stable $FE_{propanol}$ above 30% and the corresponding full-cell EE of over 16% are maintained for over 100 h with the n-propanol partial current above 300 mA ($5 cm^2$ electrode). Furthermore, operando X-ray absorption spectroscopy and theoretical studies reveal that the structurally-flexible Pb-Cu surface can adaptively stabilize the key intermediates, which strengthens the *CO binding while maintaining the C−C coupling ability, thus promoting the CO-to-n-propanol conversion.

N-propanol is a promising chemical with a diverse range of uses, e.g., acting as a feedstock in pharmaceutical industries and fuels[1–3]. Its annual global market size is forecast to reach USD 1.06 billion by 2027[2]. Nowadays, n-propanol is typically manufactured via a catalytic hydrogenation step of propionaldehyde followed by the thermal carbonylation reaction of ethylene and carbon monoxide (CO)[3]. Alternatively, it can be produced directly from electrochemical CO reduction reaction

(CORR) using renewable electricity, providing a promising way to achieve a closed carbon cycle[4–6]. Previous reports have suggested that copper (Cu) is the exclusive catalyst that can enable the coupling of the *CO intermediates to form multi-carbon ($C_{2+}$) products in CORR[7–9]. However, n-propanol synthesis on the Cu surface still suffers from a limited Faradaic efficiency (FEs < 40%), low reaction rate and poor stability[10–16], which severely prohibits its industrial applications.

[1]State Key Laboratory of Molecular Engineering of Polymers, Department of Macromolecular Science, Fudan University, Shanghai 200438, China. [2]Department of Chemistry, MOE Key Laboratory of Computational Physical Sciences, Shanghai Key Laboratory of Molecular Catalysis and Innovative Materials, Fudan University, Shanghai 200438, China. [3]National Laboratory of Solid State Microstructures, Jiangsu Key Laboratory of Artificial Functional Materials, College of Engineering and Applied Sciences and Collaborative Innovation Center of Advanced Microstructures, Nanjing University, Nanjing 210093, China. [4]Clean Nano Energy Center, State Key Laboratory of Metastable Materials Science and Technology, Yanshan University, Qinhuangdao 066004, China. [5]College of Chemistry and Chemical Engineering, Xiamen University, Xiamen 361005, China. [6]Department of Physics, University of Warwick, Coventry CV4 7AL, UK. [7]Hefei National Laboratory, Hefei 230088, China. [8]These authors contributed equally: Wenzhe Niu, Zheng Chen. ✉e-mail: xxchem@fudan.edu.cn; bozhang@fudan.edu.cn

Recent experimental and theoretical studies have demonstrated that enhancing the surface adsorbed CO (*CO) coverage and increasing the chance of $*C_2$ immediate coupling with an additional *CO provided an effective path to boost the n-propanol formation[10,11,17–19]. Introducing low-coordinated Cu sites has been reported as an effective way to increase the surface *CO coverage on catalysts[11,20–22]. However, it is still challenging to generate highly abundant and stable low-coordinated Cu sites on the catalyst surface[23–26], which prevents further improvement on the n-propanol selectivity and the long-period stability of the catalyst during CORR.

"Atomic size misfit" is a useful strategy in the field of functional materials[27–31], which often promotes the intralattice stress to induce and stabilize low-coordinated sites. Therefore, we propose that doping Cu with main group metals of large radii may induce low-coordinated sites with high abundance and stability. The radius of the lead (Pb) atom is larger than that of the Cu atom. Due to the mismatch of their atomic radii, it might form a defect-rich structure for Cu after the Pb-doping[32–35], so as to obtain rich low-coordinated sites.

Here, we developed a facile method to synthesize Pb-doped Cu (Pb-Cu) nanoparticles through electrodeposition of Pb atoms onto oxide-derived Cu (OD-Cu) surfaces as an efficient electrocatalyst for CORR to n-propanol. Structural characterizations showed that the Pb-doping could induce numerous grain boundaries (GBs) for Pb atomic enrichment. Spectroscopic studies proved that the Pb-Cu catalyst had abundant Pb-rich Cu GB sites, which effectively improved the surface coverage of *CO. The Pb-Cu catalyst achieved a CO-to-n-propanol FE of $47 \pm 3\%$ with a high half-cell energy conversion efficiency (EE) of 25%. Using a membrane electrode assembly (MEA) device, we obtained a stable CO-to-n-propanol conversation over 100 h with n-propanol FEs above 30% at a total current of 1 A with the highest full-cell EE over 18%. Theoretical calculations confirmed that the Pb-doping could promote the distortion of the Cu lattice to generate and stabilize a large number of low-coordinated Cu sites, which effectively improved the *CO coverage on the Cu surface. We also found that the Pb-doping could make the surface structure of the Pb-Cu catalyst more flexible and adaptive,

which enhanced the *CO binding while maintained the C–C coupling ability, thereby effectively promoting the carbon chain growth to form the $C_3$ product such as n-propanol.

## Results

### Catalyst preparation and characterization

The Pb-Cu catalyst was synthesized first by drip coating mix inks of the as-prepared CuO nanopowders (Supplementary Fig. 1) and Pb salts on a carbon paper, followed by an operando electroreduction process under a CO gas environment, in which Pb ions dissolved from the precatalyst and redeposited on the electrode surface, while CuO particles were electrochemically reduced to Cu (Fig. 1a). For comparison, a Cu catalyst was prepared under the same condition without Pb salts. The X-ray diffraction (XRD) pattern of the Pb-Cu electrode indicates the Cu lattice without crystalline Pb features, similar to that of the Cu electrode (Supplementary Fig. 2). The XRD diffraction peaks located at ~43.3°, ~50.4° and ~74.1° (corresponding to the (111), (200) and (220) planes of Cu) slightly shifts to the lower angle region, revealing a slightly lattice expansion of the Pb-Cu sample. A noticeable difference in morphology between the Pb-Cu and the Cu catalysts was observed via scanning electron microscopy (SEM; Supplementary Figure 2), indicating that the Pb deposition leads to the formation of small grains. Transmission electron microscopy (TEM) images confirm the topographic differences of the Pb-Cu catalyst within a single particle, as compared with the Cu catalyst (Supplementary Figs. 3a and 4a). High-resolution TEM (HRTEM) images and the corresponding selected area electron diffraction patterns reveal that the Pb-Cu surface presents denser strip structures and rich GBs, which were not observed on the Cu catalyst (Supplementary Figs. 3 and 4).

High-angle annular dark-field scanning transmission electron microscope (HAADF-STEM) images provide further insights into the microstructures of the Pb-Cu and the Cu catalysts (Fig. 1b, c, Supplementary Fig. 5). The Pb-Cu catalyst possesses an average grain size of ~20 nm and abundant GBs, which clearly contrasts with the Cu catalyst. High-resolution energy dispersive X-ray spectroscopy (EDX) elemental

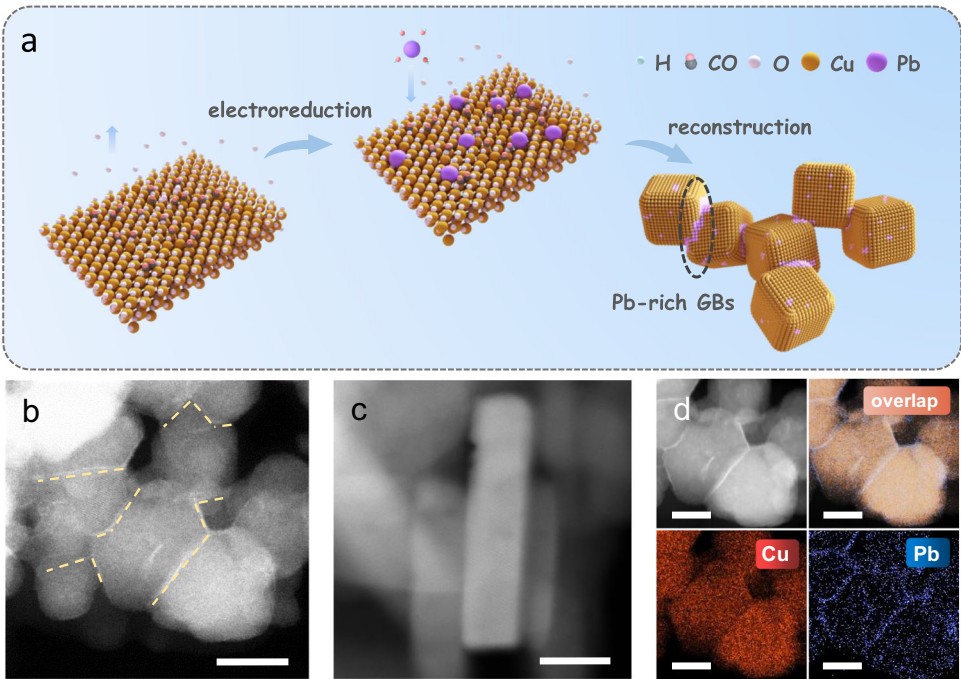

**Fig. 1 | Structural and compositional analyses of the Pb-Cu and the Cu catalysts.** **a** Scheme of the synthesis of the Pb-Cu and the Cu electrocatalysts. (**b**, **c**) High-angle annular dark-field scanning transmission electron microscopy (HAADF-STEM) images taken from the edge of nanoparticles of the Pb-Cu and the Cu samples, respectively. Scale bar, 10 nm. **d** Scanning transmission electron microscopy (STEM) images and corresponding energy-dispersive X-ray spectroscopy (EDS) elemental mapping images of Pb-Cu. Scale bar, 10 nm.

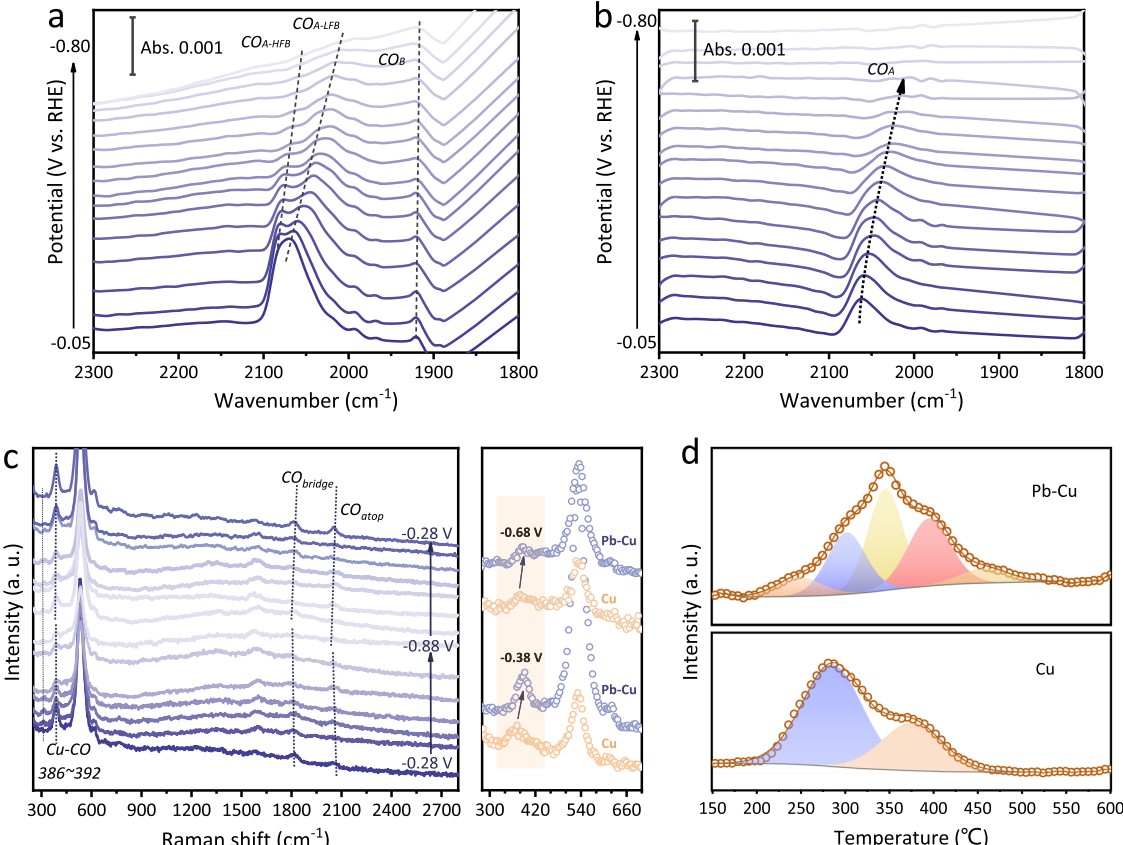

**Fig. 2 | Operando spectroscopic studies and characterization of different catalysts on CO adsorption. a, b** Operando electrochemical ATR-SEIRAS spectra of (**a**) the Pb–Cu and (**b**) the Cu catalysts under different applied potentials versus RHE during CORR. (**c**) Operando Raman spectra of the Pb-Cu catalyst under different applied potentials versus RHE during CORR. The spectra were recorded on the same point at different potentials from −0.28 to −0.88 V and backward to −0.28 V (vs. RHE). Right: Comparison of Cu−CO stretching peaks of different catalysts under different applied potentials versus RHE. The spectral intensity of the Cu catalyst is multiplied by 4. **d** CO-TPD profiles of the Pb–Cu and the Cu catalysts.

mapping demonstrates a higher dense of the Pb atoms among the GB zone of the Pb-Cu catalyst, which reveals that the Pb-doping might be the main reason of the GB formation, which induced numerous low-coordinated Cu sites. (Fig. 1d).

The average Pb atomic ratio on the Pb-Cu electrode was characterized by inductively coupled plasma optical emission spectrometer (ICP-OES) as ~2.9% (Supplementary Table 1). High-resolution X-ray photoelectron spectroscopy (XPS) further shows that the near surface Pb containing is approximately 8% (Supplementary Figs. 6, 7 and Supplementary Table 2). Hence, the as-synthesized Pb-Cu catalyst contains abundant Pb-rich Cu-GB sites.

We then performed operando X-ray absorption spectroscopy (XAS) to examine the operando formation of the Pb-Cu catalyst (Supplementary Fig. 8). Quick-scanning XAS records the Cu reduction process during the generation of the Pb-Cu and the Cu catalysts from CuO nanoparticles in the pre-catalysts. Combined with the corresponding Fourier transform spectra, we observed a quick $Cu^{2+}$ to $Cu^{0}$ transformation in 120 s on both catalysts (Supplementary Figs. 9, 10). Consequently, no trace of oxygen is resolved in the fully-derived Cu lattices. Operando Cu K-edge and Pb $L_3$-edge X-ray absorption near edge structure spectra of the Pb-Cu and the Cu electrodes further prove that Cu and Pb both remain in the metallic state during CORR (Supplementary Figs. 11 and 12). Consistent results are shown by the operando extended X-ray adsorption fine structure (EXAFS) spectra. An EXAFS fitting analysis at the Cu K-edge reveals the presence of Pb, the bond lengths of about 2.54 and 2.73 Å are attributed to the Cu-Cu and the Cu-Pb paths[36], respectively (Supplementary Fig. 13 and Supplementary Table 3). In contrast, only the Cu-Cu path was observed on

the Cu catalyst (Supplementary Fig. 14). In addition, the coordination number of the Cu-Cu path in the Pb-Cu is obviously lower than that in the Cu catalyst, revealing a low-coordinated feature of the GB-site-rich Pb-Cu catalyst. The corresponding wavelet-transform (WT)-EXAFS analysis at the Cu K-edge show that the WT maximum for the Pb-Cu sample (9.91 Å⁻¹) exhibits a higher *k*-value than that for the Cu sample (8.80 Å⁻¹), revealing that the existence of Cu-Pb interactions (Supplementary Fig. 15).

To gain insights into the CO adsorption features of the Pb-Cu surface, we performed the operando electrochemical attenuated total reflection surface-enhanced infrared absorption spectroscopy (ATR-SEIRAS) under the CORR conditions (Supplementary Fig. 16)[37,38]. Note that, CO adsorption on Pb was not considered in peak assignments, since CO was found not to adsorb on the Pb sites by the DFT calculations (Supplementary Fig. 17). The adsorption bonds in the range of 2027–2080 cm⁻¹ are attributed to the atop-bound *CO (*$CO_{atop}$) on the Cu surface (Fig. 2a, b)[37–40]. As shown in Fig. 2a, the *$CO_{atop}$ peak on the Pb-Cu surface split into two bands, which were named as the low frequency band (LFB) at ~2027–2067 cm⁻¹ and the high frequency band (HFB) at ~2075–2080 cm⁻¹. According to the literature[41,42], the LFB and the HFB are attributed to *$CO_{atop}$ on the terrace and the low-coordinated sites, respectively. We also observed bridge-bound *CO species (*$CO_{bridge}$) on the Pb-Cu surface[37,43–45]. On the contrary, the HFB of *$CO_{atop}$ and *$CO_{bridge}$ are both absent on the Cu surface. These results reveal that more low-coordinated sites on the Pb-Cu sample enable multiple *CO adsorption configurations and variation of the surface *CO coverage[46–48]. Additionally, it was found that the adsorption bands were strong under small biases, and as the potential shifts

more negatively, the *CO peaks decreased rapidly. We further provided the backward scan of the ATR-SEIRAS on both catalysts (Supplementary Fig. 18). The *CO peaks decreased as the potential scanned negatively, and increased again as the potential shifted back to a small bias. These results confirmed that the detected *CO species are the precursors for the subsequent C−C coupling processes at reaction potentials in CORR.

To further understand the CO adsorption on different electrodes, operando Raman spectroscopy measurements during CORR were performed under different potentials (Fig. 2c and Supplementary Figs. 19, 20). The bands in the range of 1900–2150 cm$^{-1}$ arise from the C≡O stretching of the *CO on the Cu sites[49], wherein the regions below and above 2000 cm$^{-1}$ are attributed to the *CO$_{bridge}$ and the *CO$_{atop}$, respectively. Consistent with the ATR-SEIRAS results, the *CO$_{bridge}$ species were only obtained on the Pb−Cu catalyst, and the *CO bands changed reversibly as the potential shifted negatively and subsequently backward. It is to note that the potential where the *CO$_{atop}$ peaks almost disappear in Raman (−0.78 V, Fig. 2c) is more negative than that in ATR-SEIRAS spectra (−0.5 V, Fig. 2a, b). This might be due to the mass transport difference between the gas diffusion electrode and a smooth Au/Si electrode[40]. Besides, the bands located at ~283 cm$^{-1}$ and ~363 cm$^{-1}$ are associated with the frustrated rotation and the stretching of *CO on Cu[14,50], respectively. We observed a blueshift of the Cu-CO stretching band on the Pb-Cu surface, as compared with that on the Cu surface (Fig. 2c), indicating a stronger bound of *CO on the Pb-Cu surface[14,50,51]. This is further confirmed by the temperature programmed desorption of CO (CO-TPD), the main desorption peak of Pb-Cu (362 °C) occurs at higher temperature than that of the Cu catalyst (294 °C) (Fig. 2d), suggesting a stronger *CO bonding on the Pb-Cu sample.

## Performance for CO-to-n-propanol electroreduction

The performance for CO-to-n-propanol conversion was evaluated in a flow-cell reactor[7–13,15]. Figure 3a shows the FEs of the multi-carbon (C$_{2+}$) products on the Pb-Cu catalyst during CORR in the potential range of −0.38 to −0.98 V (versus Reversible Hydrogen Electrode (RHE)) with 1 M KOH as the electrolyte. The production of n-propanol emerges at −0.58 V and the peak FE value of n-propanol occurs at −0.68 V as 47 ± 3%, among the state-of-the-art n-propanol-selective catalysts ever reported[10–16] (Supplementary Fig. 21 and Supplementary Table 9). In contrary, the major products on the pure Cu catalyst are C$_2$ species (ethanol, acetate and ethylene), and the highest n-propanol FE is 28 ± 2% at a more negative potential (−0.88 V) than that on the Pb-Cu catalyst (Supplementary Fig. 22). The n-propanol FEs are obviously enhanced on the Pb-Cu electrode in the range from −0.58 to −0.98 V, the n-propanol current density achieved 38 ± 2 mA cm$^{-2}$, 2-times improvement relative to that on the Cu electrode at −0.68 V (Fig. 3b, c). In addition, as compared with the pure Cu electrode, the FE ratio of n-propanol in total multi-carbon products on the Pb-Cu catalyst presents a 2-times enhancement at −0.68 V (Fig. 3d and Supplementary Fig. 23). It was noted that, in comparison with the Cu catalyst, the total FEs of the C$_{2+}$ products on the Pb-Cu catalyst are similar. We propose that the increased low-coordinated sites on the Pb-Cu catalyst have enhanced the surface *CO coverage and improved the chance for the C$_1$–C$_2$ coupling to branch the pathways from C$_2$ to the C$_3$ spices formation. The CORR performance is also evaluated on the commercial Cu nanoparticles (Com-Cu) (Supplementary Fig. 24). In comparison with Com-Cu, our Cu catalyst showed a much higher selectivity of the C$_{2+}$ products including n-propanol. This is consistent with previous reports that OD-Cu can enhance the C$_{2+}$ formation by increasing the local *CO concentration[11,12,52–54].

Partial current densities were also normalized to electrochemically active surface areas (ECSAs), which reveals an even higher intrinsic activity toward n-propanol on the Pb-Cu as compared to that on both the Cu and the Com-Cu samples (Fig. 3e, f, Supplementary

Figs. 25, 26 and Supplementary Table 4). It indicates that the enhanced CO-to-n-propanol activity on the Pb-Cu catalyst should not be attributed to the differences of particle size and surface area.

In the following, we evaluated other samples with different ratios of Pb containing. To regulate the Pb concentration, we adjusted the amount of Pb salts in the pre-catalyst inks lower and higher, denoted as Pb-Cu-l and Pb-Cu-h, respectively. Both Pb-Cu-l and Pb-Cu-h retained similar morphology and structure as the Pb-Cu catalyst (Supplementary Figs. 27–31). The bulk atomic percentages of Pb in the Pb-Cu-l and the Pb-Cu-h catalysts were tuned to 0.9% and 3.6%, respectively, based on ICP-OES (Supplementary Table 1). The Pb atomic percentages were also detected by XPS as 2.3% for Pb-Cu-l and as 17.0% for Pb-Cu-h (Supplementary Fig. 32 and Supplementary Table 2). We found that both Pb-Cu-l and Pb-Cu-h catalysts exhibit enhanced FEs of n-propanol in the potential range of −0.68 to −0.88 V (Supplementary Figs. 33, 34), as compared with the bare Cu sample, and the best n-propanol performance was obtained on the Pb-Cu catalyst with ~8% Pb atoms detected by XPS.

To further understand the relationship between the n-propanol selectivity and the surface *CO adsorption, we also performed the operando electrochemical ATR-SEIRAS on the Com-Cu (Supplementary Fig. 35) where the *CO coverage is lower than that on the Cu sample under the CORR condition. It is obvious that the n-propanol FEs on our three prepared samples are directly related to the surface *CO coverage. Furthermore, we reduced the CO partial pressure by gas mixtures of CO and Ar with a composition of 75%, 50%, 25%, 10% and 5%, respectively. With the decreasing of the surface *CO coverage, the normalized FE[17] (the selectivity among only carbon-base products by excluding the H$_2$ contribution) of n-propanol on the Pb-Cu decreased from ~48 to ~21% with the ethylene FE increased from ~18 to ~49% (Supplementary Fig. 36). Besides, the n-propanol FE on the Pb-Cu under 10% CO is similar to the value on Com-Cu under 100% CO at the same potential. These prove that the abundant Pb-rich low-coordinated sites improve the n-propanol production mainly via enhancing the surface *CO concentration. We also assessed the proton concentration dependency on the n-propanol formation. The highest n-propanol FE was obtained in the 1 M KOH, in the range of 0.5–3 M KOH (Supplementary Fig. 37). Further increasing of the solution pH prioritizes the acetate formation due to the increased tendency for the OH$^-$ attack towards the C$_2$ intermediates[8,55].

The high FEs (≥ 40%) of n-propanol on the Pb-Cu electrode exhibit a 10 h-stability in a home-made flow-cell reactor (Supplementary Fig. 38). The stability of the Pb-Cu catalyst is much better than that of the Cu catalyst, which might be attributed to the high stability of low-coordinated Cu sites after the Pb-doping. Additionally, we further performed a stability evaluation of the Pb-Cu catalyst on a 5 cm², two-electrode MEA electrolyzer. At total current of 1 A, the highest n-propanol FE and the full-cell EE of 38% and 18% were obtained, respectively. The n-propanol partial current reaches 378 mA with a current density of 76 mA cm$^{-2}$. The catalytic performance in an MEA electrolyzer is not totally in line with that in flow-cell. As a high n-propanol FE was also obtained in an MEA electrolyzer, we supposed that the Pb-Cu catalysts still exhibit a good ability of enhancing surface *CO coverage to promote the C$_3$ formation as that in flow-cell. Moreover, the n-propanol FEs above 30 % and the full-cell EE over 16% can be maintained for over 100 h with a suppressed competitive reaction of H$_2$ evolution (FE$_{H2}$ < 5%) (Fig. 3g and Supplementary Figs. 39, 40). TEM and XPS analysis on the post-reaction catalysts reveal that the Pb-Cu catalyst retains its structure after a long-period operation, which further confirms the role of Pb in stabilizing the low-coordinated Cu sites (Supplementary Figs. 41, 42).

## Theoretical calculations

Theoretical modeling was performed to gain an atomic insight of the low-coordinated Cu sites induced by the Pb-doping and to confirm the

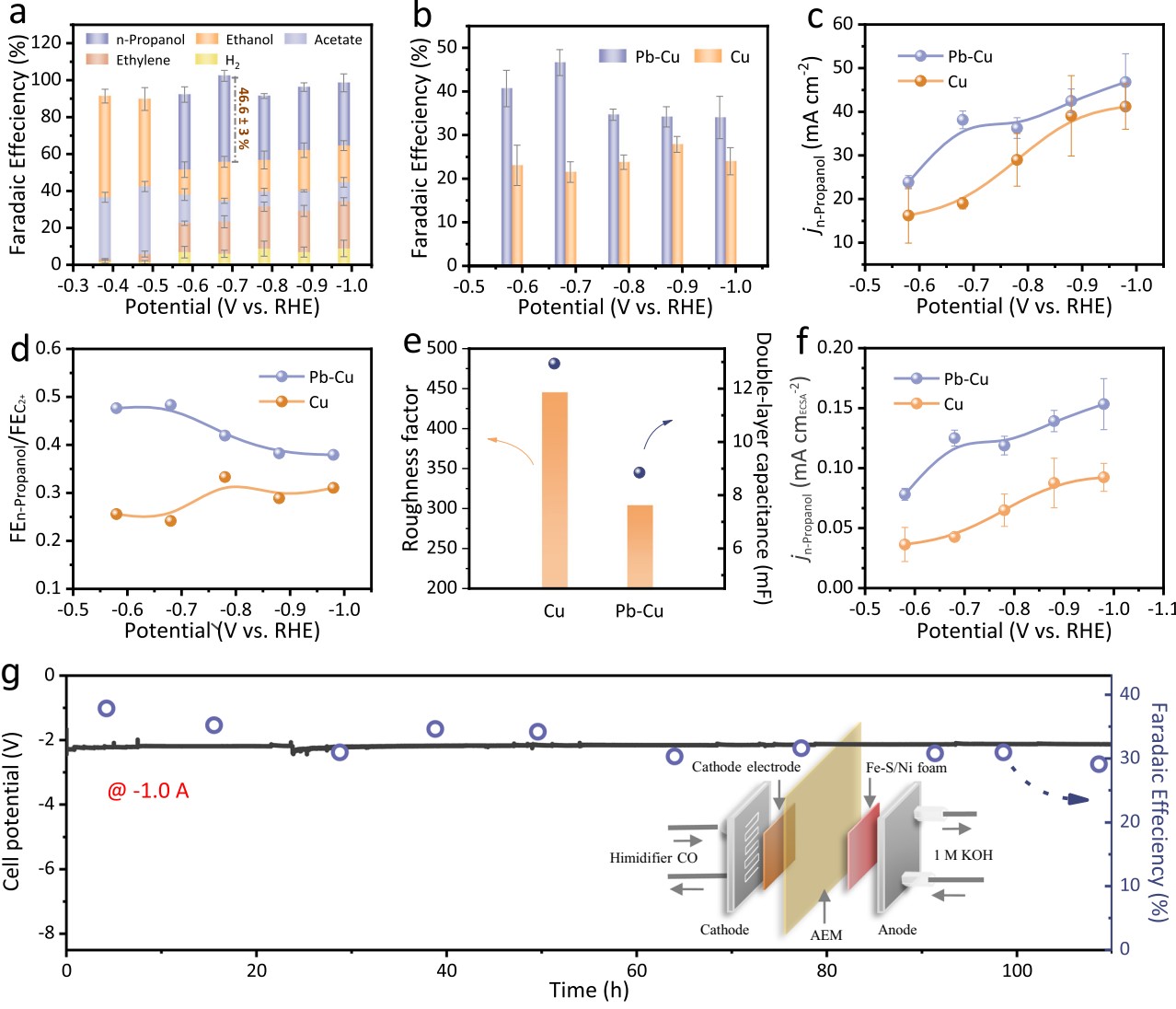

**Fig. 3 | CORR performance of different cathode electrodes. a** CORR products distribution under different potentials for the Pb-Cu and the Cu electrodes. **b** FE$_{n-propanol}$ on different electrodes at various potentials. **c** Partial n-propanol current densities on different electrodes at various potentials. **d** Comparison of FE$_{n-propanol}$/FE$_{C2+}$ ratios on different electrodes at various potentials. **e** Electric double-layer capacitances and surface roughness factors of catalysts calculated by defining the surface roughness factor for electropolished polycrystalline Cu with an electric double layer capacitance of 29 mF as 1 on different electrodes[8]. **f** Electrochemically active surface area (ECSA)-normalized partial n-propanol current densities on different electrodes. Error bars in (**a**), (**b**), (**c**) and (**f**) represent the standard deviation of at least three independent samples. **g** FE$_{n-propanol}$ and cell voltage during 110 h operation of CORR at a constant current of 1.0 A. CO feed rate is 40.0 ml min$^{-1}$. Inset, schematic diagram of the MEA system. Source data in (**a**), (**c**) and (**g**) are provided as a Source Data file.

experimental observation that *CO tends to be concentrated by such a structure. Furthermore, the possibility for the Pb-rich Cu-GB sites to stabilize the key intermediates of both C$_2$ (*COCOH)[56,57] and C$_3$ (*COCOHCO, *COCCH$_2$)[11,58] was investigated theoretically so as to understand the high C$_3$ selectivity over the Pb-Cu surface.

The as-synthesized Pb-Cu surface is not expected to have an ordered structure. It is, thus, quite challenging to determine its atomic structure in experiment, especially under the working condition. Herein, we performed the ab-initio molecular dynamic (AIMD) simulation at high temperature (600 K) to explore the possible atomic structures of the Pb-Cu surface. The prevailing geometries were picked out from the MD trajectory (Supplementary Fig. 43), which were then optimized in normal DFT calculations (Supplementary Fig. 44). As enlightened by the characterization results that Pb atoms prefer GBs more than the terrace surface, the modeling was started with the Pb (10.4%) doped Cu(211) surface, which exposes low-coordinated Cu sites similar to GBs. Since the catalyst was synthesized and was working under the CO atmosphere, it is important to consider the *CO coverage

effects on both Pb-doped and undoped Cu(211) surfaces. According to the observation that only on-top *CO was found on the undoped Cu (Fig. 2b), a 0.5 ML *CO coverage was employed to consider the high *CO coverage on the Cu(211) surface, where the bridge *CO would appear at coverage larger than 0.5 ML[59]. For comparison purpose, the same *CO coverage was used in the Pb doped Cu(211) surface. More details about DFT calculations, AIMD simulations and the Pb-Cu surface model were provided in the Supplementary Information.

While Supplementary Fig. 43 indicates the conformations of the Pb-Cu surface explored by the AIMD simulations, Supplementary Fig. 44 shows some selected stable candidates after DFT optimizations. As shown by these structures, the Pb doped Cu(211) surface was significantly reconstructed (Fig. 4a and Supplementary Figs. 43, 44), which became more open with less surface atoms in the same area than that on the Cu(211) surface, leading to a higher percentage of low-coordinated Cu sites (highlighted by the balls in gold). These theoretical results lend strong support to the arguments from the experiment that Pb induces and stabilizes low-coordinated Cu sites like GBs, which

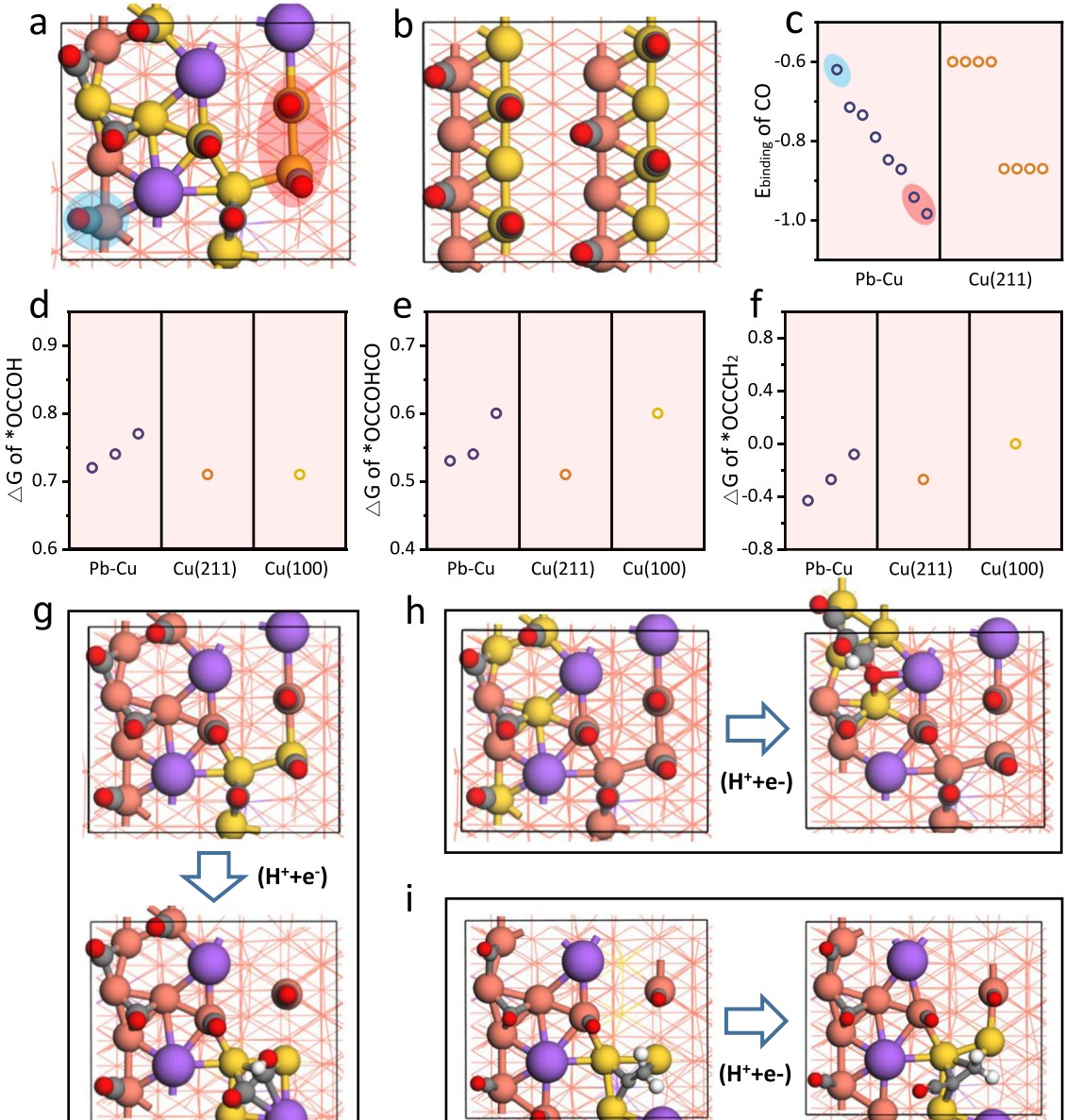

**Fig. 4 | DFT calculations.** The structures of (**a**) the Pb-Cu surface and (**b**) the Cu(211) surface at a high *CO coverage. **c** The binding energy of *CO on (**a**) the Pb-Cu surface and (**b**) the Cu(211) surface at a high *CO coverage. As the Pb-doping induced many different low-coordinated Cu atoms, the *CO binding energies of eight CO in (**a**) are all different, while those in (**b**) only show two different types of binding energies corresponding to the flat and stepped sites, respectively. The binding energies for each surface are displayed in the decreased order. Among them, *CO (within red circle) on Cu with lowest coordination number shows the strongest binding energies while that on Cu with the highest coordination number (within blue circle) shows the weakest binding energy. The formation free energy of (**d**) *COCOH, (**e**) *COCOHCO and (**f**) *COCCH$_2$ on the Pb-Cu surface and the Cu(211) surface at a high *CO coverage, and the Cu(100) surface at a low *CO coverage. This is consistent with the results of ATR-SEIRAS. The C−C coupling on three different sites of the Pb-Cu surface were calculated and displayed (black points) in activity decreased order in (**d**), (**e**) and (**f**), and compared to those on stepped site of the Cu(211) surface (orange points) and flat site of the Cu(100) surface (yellow points). The most active examples for (**g**) *COCOH, (**h**) *COCOHCO and (**i**) *COCCH$_2$ formation on the Pb-Cu surface. All the others can be found in the Supplementary Figs. 45−47. The orange, purple, gray, red and white balls present Cu, Pb, C, O and H respectively. The golden balls present the highlighted Cu atoms. The low-coordinated (<=7) Cu atoms are highlighted in (**a**) and (**b**). The Cu atoms where the C−C coupling happens are highlighted in (**g**), (**h**) and (**i**).

also supported the flow-cell stability test results in experiment that the Pb doped Cu is much more stable than the undoped Cu (Supplementary Fig. 38).

We then compared the binding strength of *CO on the Pb-Cu surface (Fig. 4a) and that on the Cu(211) surface (Fig. 4b). As shown in Fig. 4c, there are two types of *CO bindings on the Cu(211) surface, which correspond to *CO on the step sites and *CO on the terrace sites, respectively. On the Pb-Cu surface, there are various types of *CO bindings, which are, in general, stronger than those on the Cu(211) surface. This can be attributed to the fact that the low-coordinated Cu

atoms were more dispersed over the Pb-Cu surface (golden balls), while those (golden balls) on the Cu(211) surface are concentrated at the step sites, and hence are difficult to be occupied simultaneously because of the repulsion between *CO. These results are also consistent with the TPD results (Fig. 2d), where the Pb doped Cu showed more *CO desorption peaks with stronger *CO binding strength than the undoped Cu. The stronger *CO binding strength at a high coverage on the Pb-Cu surface is helpful to maintain the desired high *CO coverage under the working conditions as indicated by the operando spectroscopy (Fig. 2).

Generally, the stronger *CO binding strength would lead to a high *CO coverage, it would have made the CO–CO couplings more difficult[60,61]. Therefore, we compared the formation energies of the key intermediates for $C_2$(*COCOH) and $C_3$(*COCOHCO) products on the Pb-Cu surface to those on the Cu(211) and Cu(100) surfaces. Unlike those on the Pb-Cu surface (Fig. 4a) and the Cu(211) surface (Fig. 4b), calculations on the Cu(100) surface were performed at low coverage, which is in consistent with the results of ATR-SEIRAS (Fig. 2a, b and Supplementary Fig. 35). However, the results showed that the formation energies for both *COCOH (Fig. 4d) and *COCOHCO (Fig. 4e) on the Pb-Cu surface are comparable to those on the Cu(211) and Cu(100) surfaces, despite a stronger *CO binding strength for the former. In addition, we also studied the coupling reaction of *CO and *CCH$_2$, which was also suggested as a possible source of the $C_2$ precursor. Similar to COCOHCO, the reaction free energies (Fig. 4f) on two sites of the Pb-Cu surface are comparable to that on the Cu(211) surface, while a higher reaction energy on one site of the Pb-Cu surface is comparable to that on the Cu(100) surface. It was found that the formation of *COCOH tends to induce a square-like site-assembly (Fig. 4f and Supplementary Fig. 45), while the formation of *COCOHCO and *COCCH$_2$ tend to induce a long and narrow site-assembly (Fig. 4h, i and Supplementary Figs. 46, 47). This demonstrates that the Pb-Cu surface can adapt itself to stabilize intermediates with different shapes, which indicates that the Pb-doping makes the surface structure more flexible and adaptive. This structural flexibility echoes the results from the operando EXAFS and the corresponding fitting analysis under different potentials using the Pb-Cu and the Cu catalysts[36] (Supplementary Figs. 48, 49 and Supplementary Tables 7–8). Hence, the coordination numbers of the Cu–Cu path and the Cu-Pb path exhibit reversible changes with the potential shift on the Pb-Cu sample, revealing a variable coordination structure of the Pb-Cu surface. On the contrary, the Cu catalyst shows a stable structure, whose coordination number remains at 12 under different potentials during CORR. Therefore, both the experiments and the calculations suggest a structurally flexible and adaptive feature of the Pb-Cu surface. Benefiting from this feature, the Pb-Cu catalyst can obtain a stronger *CO bonding for the increased *CO coverage, while at the same time the ability of the CO-CO couplings for the carbon chain growth (Fig. 4c, d). Indeed, the increased *CO coverage, in turn, can accelerate the rate of the carbon chain growth, which is in proportion to the square or cubic of $\theta_{CO}$ determined by the number of *CO in the corresponding rate-determining step[62,63]. As compared to OD-Cu, the ~2 times higher n-propanol selectivity of Pb-Cu catalyst (Supplementary Fig. 22) is more likely due to the higher $\theta_{CO}$, whereas an obvious energy differences would result in reactivity differences in orders of magnitude. This conclusion is also supported by the experimental results that the n-propanol selectivity on the Pb-Cu catalysts highly depends on the CO concentration (Supplementary Fig. 36).

In summary, we report a Pb-doped Cu catalyst with abundant low-coordinated Cu sites induced by the atomic size misfit of Pb over Cu. It enables a high-efficiency CO-to-n-propanol conversion with FE of $47 \pm 3\%$ at a partial current density of ~35 mA cm$^{-2}$. We also obtain a 100-h stable n-propanol generation on the Pb-Cu electrode based the MEA electrolyzer during CORR. Combined the results from experiments and theoretical calculations, we attribute this high n-propanol selectivity to the abundant Pb-rich Cu-GB sites, which permit a strong CO adsorption with various configurations to effectively achieve a high surface *CO coverage. Moreover, we find that the Pb-Cu surface is flexible and self-adaptable during the reactions, such that the catalyst also maintains a good C–C coupling ability. It is the unique feature of the as-synthesized Pb-Cu catalyst that promotes the formation of the $C_3$ selectivity. The finding in this work may shed new light on the rational catalyst design for CO-to-n-propanol electrosynthesis and the industry utilization of $CO_2$RR and CORR.

## Methods

### Preparation of CuO nanopowders

CuO nanopowders were fabricated by a solvothermal method. Copper chloride dehydrate (1.02 g, 6.0 mmol) and 50 mg Nano-carbon black were first dissolved in 30 ml 2 M sodium hydroxide solution, under stirring for 30 min at room temperature. Then the mixture was transferred into a Teflon-lined stainless-steel autoclave (50 mL) for solvothermal treatment at 130 °C for 12 h, and subsequently centrifuged three times with deionized water and ethanol, respectively. The sample was finally dried in a vacuum oven at 80 °C for 6 h.

### Preparation of gas diffusion electrodes (GDEs)

As for GDE with the Pb-Cu pre-catalyst, a catalyst slurry of 15 mg CuO nanopowders, 3 mg Pb(NO$_3$)$_2$, 1 mL methanol and 50 μL of Nafion solution was mixed and sonicated. Then, the slurry was drip-coating on a (2 cm × 2 cm) GDL, the as-prepared GDL was annealed in a tube furnace under 200 °C for 2 h. As for Pb-Cu-l and Pb-Cu-h, to regulate the Pb concentration, we adjusted the amount of the Pb salts in pre-catalyst inks to 1 mg and 6 mg, respectively. Electrodes with the Cu pre-catalyst were prepared by a similar procedure, 15 mg CuO nanopowders, 1 mL methanol and 50 μL of Nafion solution was mixed to form a catalyst slurry and sonicated. The subsequent procedures for the preparation were the same as those for the preparation of GDE with the Pb-Cu pre-catalyst.

The as-prepared GDEs were run for 300 s at −0.38 V (vs. RHE) in 1 M KOH, until a stable current has been gained to make sure the pre-catalysts have been completely electro-reduced to a stable state as the CORR catalysts.

The electrode potentials were rescaled to the RHE reference by the following equation:

$$E(vs.\ RHE) = E(vs.\ Hg/HgO) + 0.098V + 0.0591 \times pH \qquad (S1)$$

### Structural characterization

The morphologies of these catalysts were acquired using a Hitachi FE-SEM S-4800 SEM operated at 1.0 kV. High-resolution transmission electron microscopy (HRTEM) images were taken on a JEOL JEM-2100F TEM operated at 200 kV. Scanning transmission electron microscopy (STEM) was carried out on FEI Titan Cubed 60–300 at an accelerating voltage of 300 kV and JEOL ARM-200F equipped with a cold field emission gun and a Cs corrector (CEOS) for probing lenses at the operation voltage of 200 kV. High-resolution EDX was based on super-X detector. XPS measurements were carried out on PHI 5700 ESCA System using Al Kα X-ray radiation (1486.6 eV) for excitation. Powder XRD patterns were obtained with a MiniFlex600 instrument in Bragg-Brettano mode using 0.02° divergence with a scan rate of 0.1° s$^{-1}$.

### Operando X-ray absorption fine spectroscopy (XAFS)

The operando Cu K-edge XAFS measurements were performed on the 1W1B beamline of the Beijing Synchrotron Radiation Facility, China. The Cu K-edge quick X-ray absorption fine structure (QXAFS) data were recorded from 8.8 to 9.2 keV in fluorescence mode with a step size of 0.5 eV at the near edge. About 40 s were consumed for each QXAFS spectrum (including 30 s to collect the data and 10 s to reset the detector position).

The Pb-Cu GDE (same as the electrochemical measurements) was carried out with a chronoamperometry process at −0.68 V (vs. RHE) in a home-made flow-cell type reactor for the operando XAS measurements, similar to the flow cell used for preference measurements, and the only difference is that the outer surface of the gas chamber was replaced with the Kapton tape (Supplementary Fig. 8). In the flow cell reactor, Hg/HgO reference electrode (1 M KOH), Fe-S/ Ni foam electrode and anion exchange membrane (Fumatech FAB-PK-130) were used as the reference electrode, anode, and membrane, respectively.

1 M KOH aqueous solution was used as the electrolyte and CO (Air France, 99.99%) was continuously supplied to the gas chamber during CORR. XAS data were processed using Athena and Artemis software included in a standard IFEFFIT package. As reference samples, ex-situ Cu K-edge XAFS data of commercial Cu NPs and CuO powders was conducted. These power samples were prepared by uniformly placing powders on a piece of 3 M tape.

## Operando attenuated total reflection surface-enhanced infrared absorption spectroscopy (ATR-SEIRAS)

ATR-SEIRAS measurements were performed with a Nicolet iS50 infrared spectrophotometer with a built-in mercury cadmium telluride (MCT) detector. Different Cu-based catalysts coated on the Au/Si substrate was used as the working electrode, a Hg/HgO electrode and a graphite rod were applied as the reference and counter electrodes (Supplementary Fig. 16), respectively.

Au/Si substrate was prepared as follows: the hemicylindrical Si prism was purchased from IRUBIS GmbH. The Au film on the reflecting plane of Si prism (Au/Si substrate) was prepared according to the so-called 'two-step wet process'. Firstly, the reflecting plane of Si prism was mechanically polished with 1.0 μm, 0.3 μm and 0.05 μm $Al_2O_3$ powder, sonicated in acetone and water respectively, soaked in piranha solution and thoroughly rinsed with Milli-Q water (18.2 MΩ·cm). Then the total reflecting plane was immersed in a 40% $NH_4F$ solution for 1.5 min to terminate the Si surface with hydrogen, and was immersed in the plating solution containing 0.015 M $HAuCl_4$, 0.15 M $Na_2SO_3$, 0.05 M $Na_2S_2O_3$, and 0.05 M $NH_4Cl$ at 60 °C for 3 min to deposit Au film.

The ATR-SEIRAS spectra were acquired at a resolution of 4 cm$^{-1}$ with unpolarized IR radiation at an incidence angle of ca. 70°. The electrolyte was 0.1 M KOH, which was saturated with CO or purged with Ar gas during the experiment. The electrode potential was held at an open circuit potential (OCP) and a background spectrum was recorded. All of the spectra are shown in the absorbance unit as -log (I/I$_0$), where I and I$_0$ represent the intensities of the reflected radiation of the sample and background spectrum, respectively. The electrode potential was altered from 0.05 to −0.80 V vs. RHE in a stepwise manner. Concurrently, the infrared spectra were recorded with a time resolution of 30 s per spectrum at a spectral resolution of 4 cm$^{-1}$.

The electrode potentials were rescaled to the RHE reference by the following equation:

$$E(vs. \text{ RHE}) = E(vs. \text{ Hg/HgO}) + 0.098V + 0.0591 \times pH \qquad (S2)$$

To make our mechanistic insights based on ATR-SEIRAS more convincing, the catalytic performance was also evaluated in the ATR-SEIRAS cell. We achieved a peak n-propanol FE of ~37% at −0.68 V (vs. RHE), ~2 times higher n-propanol selectivity than that of the Cu catalysts (17%). In addition, in the potential range of −0.58 V to −0.78 V, the Pb-Cu catalysts showed much enhanced n-propanol selectivity compared with the Cu catalysts. Furthermore, both of the n-propanol and ethylene FEs were promoted and the ethanol was suppressed after the Pb doping, similar to the performance obtained in the flow-cell reactor (Supplementary Fig. 50).

## Operando surface-enhanced Raman spectroscopy (SERS)

The operando SERS measurements were performed using a Horiba Scientific Xplora Raman Microscope in a modified flow cell and a water immersion objective (100×) with a 633 nm laser (NA = 1.0, WD = 2.0 mm; LUMPLFLN-60X/W; Olympus Inc.; Waltham, MA). Each spectrum was acquired using a 5-s integration and an averaged 10 scans. The spectra were recorded and processed using the LabSpec 6.0 software. The same working electrode prepared for the electrochemical performance testing was applied for operando Raman analysis. An

Ag/AgCl electrode (3 M KCl) and a graphite rod were used as the reference electrode and the counter electrode (Supplementary Fig. 19), respectively. A 1 M KOH aqueous solution was used as the electrolyte. CO gas feedstocks were continuously supplied to the gas chamber during the measurement.

The potentials in Raman measurements were converted to values with reference to RHE using the equation:

$$E(vs. \text{ RHE}) = E\left(vs. \frac{\text{Ag}}{\text{AgCl}}\right) + 0.197 \text{ V} + 0.0591 \times pH \qquad (S3)$$

The catalytic performance was also evaluated in the Raman cell. In the potential range from −0.58 to −0.78 V, the peak n-propanol FE of ~43% was obtained at −0.68 V (vs. RHE) on the Pb-Cu catalysts, ~2 times higher n-propanol than that on the Cu catalysts. Besides, both of the n-propanol and ethylene FEs were promoted and the ethanol was suppressed after the Pb doping, similar to the performance in the flow-cell reactor and ATR-SEIRS cell (Supplementary Fig. 51).

## Temperature programed desorption (TPD)

The GDEs (same as the electrochemical measurements) were carried out with a chronoamperometry process at −0.68 V (vs. RHE) and dried in a vacuum oven. Then, the samples (including catalysts and GDLs) were grinded into powder for CO desorption measurements. CO desorption measurement of grinded GDL with the same parameter was also carried out to exclude the contribution of the GDL support.

The CO adsorption study was carried out using temperature-programed desorption instrument equipped with a thermal conductive detector (AutoChem II 2920). The catalysts were degassed under 100 °C with continuous Ar flow to remove the adsorbed gases on catalysts surface. After 1 h degassed process, the CO gas was introduced to allow sufficient adsorption of CO on the catalysts. The rest CO was swept using Ar. The temperature programed was started with continuous Ar in constant velocity to bring the desorbed CO to the detector.

## Electrochemical measurements

Without specification, the CORR performance of various catalysts was measured at 25 °C, in a flow cell configuration consisting of a gas chamber, a cathodic chamber, and an anodic chamber. The as-prepared working electrode was fixed between the gas and cathodic chambers, with the catalyst layer side facing the cathodic chamber (geometric active surface area of 1 cm$^2$). The Fe-S/Ni foam electrode and the Hg/HgO electrode (with 1 M KOH as the filling solution) were employed as counter and reference electrodes. An anion exchange membrane (AEM) (Fumatech FAB-PK-130) was used to separate the cathode and the anode chambers.

The combined catalyst and diffusion layer, anion exchange membrane and nickel anode were then positioned and clamped together using polytetrafluoroethylene (PTFE) spacers such that alkaline electrolytes could be introduced into the chambers between the anode and membrane as well as the membrane and the cathode at 10 mL·min$^{-1}$ using a peristaltic pump. The CO (Air France, 99.99%) flow was kept constant at 20 mL·min$^{-1}$ using Alicat Scientific mass flow controller and then supplied to the gas chamber. The actual flow rate was determined using a bubble flowmeter at the outlet of cathodic chamber.

The MEA electrolysis was conducted at 25 °C, in a home-made 5 cm$^2$ CO electrolyzer. An as-prepared gas-diffusion electrode (2.0 cm × 2.5 cm) was employed as the cathode, and a PTFE insulator sheet with a 5 cm$^2$ window was attached to the cathode to avoid short circuit. A pre-treated Sustainion membrane (X37-FA) and a Fe-S/Ni foam electrode (2.0 cm × 2.5 cm) were put on the top of the

membrane. Then, 1 M KOH aqueous solution was used as the anolyte and circulated using a pump at a rate of 30 ml min⁻¹. On the cathode side, CO gas (40 mL·min⁻¹) was continuously humidified with DI water and fed into the cathode chamber. The gas products were collected and tested by an in-line gas chromatograph equipped with a cold trap. Due to the liquid product crossover, the FEs of liquid products were calculated using the total amount of the products collected at anodes and cathodes.

The Fe-S/Ni foam electrodes were fabricated by a solvothermal method. Typically, Fe-S/Ni foam (Shenzhen Poxon Machinery Technology Co. Ltd., surface density: 350 g m⁻², thickness: 1.5 mm, size: 3.0 × 4.0 cm²) was ultrasonicated in acid, acetone and ethanol. Ferric chloride hexahydrate (4.87 g) and sodium sulfide nonahydrate (7.21 g) were first dissolved in 300 ml deionized water, under stirring for 2 h at room temperature. Then 35 ml mixture and Ni foam after pretreatment were transferred into a Teflon-lined stainless-steel autoclave (50 mL) for a solvothermal treatment at 150 °C for 13.5 h, and subsequently centrifuged three times with deionized water. The sample was final dried in a vacuum oven at 60 °C for 12 h.

All CO reduction experiments were performed using an electro-chemical workstation (Autolab PGSTAT302N) equipped with a 10 A current booster. The reactions were run for at least 300 s before the products were collected for analysis to make sure the pre-catalysts have been completely electro-reduced to a stable state as the CORR catalysts. Gas chromatograph (Agilent Technologies 7890B or Shang-hai Ramiin GC 2060) equipped with thermal conductivity (TCD) and flame ionization (FID) detectors were used to determine the gaseous products, which were collected from both the outlet of gas chamber and cathode chamber to make the gas test more accurate. The liquid products were analyzed offline using ¹H nuclear magnetic resonance (NMR) analysis (AVANCE III HD 400 MHz). Dimethyl sulfoxide (Sigma, 99.99%) was added as an internal standard for NMR analysis. The one-dimensional ¹H spectrum was measured with water suppression using a pre-saturation method. The Faradaic efficiencies (FEs) of liquid products were calculated based on the total amount of the products collected in anode and cathode sides during the same period due to the liquid crossover. After obtaining the n-propanol concentration of each sample from NMR quantification, $FE_{n\text{-propanol}}$ was calculated based on the following equation:

$$FE_{n-propanol} = \frac{96485 \times 4 \times \text{moles of n} - \text{propanol}}{\int i \, \mathrm{d}t} \quad (S4)$$

where $i$ is the stabilized total current during electrolysis measurements.

## Electrochemical active surface area (ECSA) calculation

The ECSAs of catalysts were calculated based on their electrical double layer capacitor ($C_{dl}$), which were obtained from CV plots in a narrow non-Faradaic potential window from 0.14 to 0.20 V (vs. RHE). The measured capacitive current densities at 0.17 V were plotted as a function of scan rate and the slope of the linear fit was calculated as $C_{dl}$. The specific capacitance was found to be 29 μF cm⁻², and the ECSA of the catalyst is calculated from the following equation:

$$ECSA = \frac{C_{dl}}{29\,\mu F\,cm^{-2}}\,cm^2 \quad (S5)$$

The intrinsic activity was revealed by normalizing the current to the ECSA to exclude the effect of surface area on catalytic performance. The ECSA values of the catalysts are listed in Supplementary Table 4.

## Cathodic energy efficiency (EE) calculation

Cathodic EE is calculated assuming the overpotential of anodic oxygen evolution reaction to be zero, which is calculated as follows:

$$n-propanol\,EE_{half-cell} = \frac{\left(1.23 + \left(-E_{n-propanol}\right)\right) \times FE_{n-propanol}}{1.23 + (-E)} \quad (S6)$$

where E is the applied potential; $FE_{n\text{-propanol}}$ is the measured Faradaic efficiency of n-propanol; $E_{n\text{-propanol}}$ is the thermodynamic potential of the CO-to-n-propanol process, i.e., 0.20 V. This potential is presented without iR correction, except in Supplementary Table 9, where a 70% iR correction was performed. The uncompensated solution resistances ($R_\Omega$) were measured by extrapolating the electrochemical impedance semi-circle to the high-frequency end, which was ca. 3.5 Ω for each electrode in 1 M KOH.

## Full-cell EE calculation

Similar to cathodic energy EE, full-cell EE is calculated as follows:

$$n-propanol\,EE_{full-cell} = \frac{\left(1.23 + \left(-E_{n-propanol}\right)\right) \times FE_{n-propanol}}{E_{cell}} \quad (S7)$$

where $E_{cell}$ is the measured cell voltage at a given current density, $FE_{n\text{-propanol}}$ is the measured Faradaic efficiency of n-propanol; $E_{n\text{-propanol}}$ is the thermodynamic potential of the CO-to-propanol process, i.e., 0.20 V.

## DFT calculations

The Vienna ab initio simulation package (VASP)[64–66] was used for all density functional theory (DFT) calculations. The $1s$ electron in H, the $2s$, $2p$ electrons in C and O, the $3d$, $4p$ electrons in Cu, and the $6s$, $6p$ electrons in Pb were treated as valence electrons, while the kinetic energy cutoff for the plane-wave basis sets was set to be 400 eV. The remaining core electrons were described by the projector augmented-wave (PAW) method[67]. The Monkhorst–Pack meshes[68] of $2 \times 2 \times 1$ k-point sampling in the Brillouin zone were employed for the slab model. For the pristine Cu(100), a $4 \times 4$ supercell (10.2 Å × 10.2 Å) consisting of 2 fixed bottom layers and 2 relaxed top layers was used. For the pristine Cu(211) and Pb-doped Cu(211) surface, a $2 \times 4$ supercell (12.5 Å × 10.2 Å) consisting of 6 fixed bottom layers and 6 relaxed top layers was used. When the convergence criterion for optimizations was met, the largest remaining force on each atom was less than 0.03 eV Å⁻¹. For all calculations, the generalized gradient approximation (GGA) of the Perdew–Burke–Ernzerhof (PBE) functional was used[69].

For CO reduction mechanism, there were proton-coupled electron transfer (PCET) steps. The Gibbs free energy change ($\Delta G$) was calculated by using the standard hydrogen electrode (SHE) model[70,71], which used one-half of the chemical potential of hydrogen as the chemical potential of the proton-electron pair. According to this method[70,71], the $\Delta G$ value was determined as:

$$\Delta G = \Delta H - T\Delta S + \Delta G_U + \Delta G_{pH}, \quad (S8)$$

where $\Delta H$ and $\Delta S$ were the enthalpy change and the entropy change, respectively. $\Delta G_U$ was the free energy contribution related to the electrode potential $U$. $T$ is the absolute temperature. $\Delta G_{pH}$ was the concentration correction to the H⁺ free energy, which was calculated as

$$\Delta G_{pH} = 2.303 \times k_B \times pH, \quad (S9)$$

where $k_B$ is the Boltzmann constant. As the theoretical overpotential was independent of the pH or the potential value $U$[72], the analysis for the free energy changes was performed at standard conditions (pH = 0, $T$ = 298.15 K, 1 atm) and $U$ = 0. During calculations, for convenience, we assumed the chemical potential of the water in solution was equal to 3.169 kPa, the same as pure liquid water at room temperature.

We assumed that in addition to the total electronic energies, the translation and rotation contributions of the gas phase were significant while other parts were ignored. Assuming the gas phase to be an ideal gas, the partition functions of translation $Q^{trans}$ and rotation $Q^{rot}$ were calculated as[73]:

$$Q^{trans} = \left(\frac{2\pi m k_B T}{h^2}\right)^{\frac{3}{2}} V, \qquad (S10)$$

$$Q^{rot} = \frac{1}{\sigma}\frac{k_B T}{hB^{rot}}(\text{linear}), Q^{rot} = \frac{1}{\sigma}\left(\frac{k_B T}{h}\right)^{\frac{3}{2}}\sqrt{\frac{\pi}{A^{rot}B^{rot}C^{rot}}}(\text{nonlinear}), \qquad (S11)$$

where $P$ and $m$ are the pressure and the molecular mass, respectively, $k_B$ is the Boltzmann constant, and $T$ (298.15 K) is the absolute temperature. $V = \frac{k_B T}{P}$ is the volume of the system, σ is the symmetry factor, $A^{rot}$, $B^{rot}$, $C^{rot}$ are rotational constants, and $h$ is the Plank's constant.

Ab initio molecular dynamics (AIMD) simulations based on Born-Oppenheimer approximation were also performed using VASP[64–66]. A time step of 2 fs was used. Canonical (NVT) ensemble and Nosé-Hoover thermostats[74,75] were set to 600 K. In the present work, the MD simulation was only used to explore the possible stable structures for the Pb-Cu surface instead of obtaining free energy. Therefore, the temperature was not necessary to be set at room temperature where experiment was performed. The relatively high temperature MD simulation was employed because the trajectory was easy to trap in some local minimum at low temperature, which was not ideal for searching the more stable structures. After MD simulations, the low energy structures in the trajectory were picked out and optimized by the normal DFT calculations described above.

## Data availability
The authors declare that all data supporting the results of this study are available within the paper and its supplementary information files or from the corresponding author upon request. The electrochemical data of CORR performances, including long-period stability data is provided as the Source Data in this paper.

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

## Acknowledgements

This work was supported by NSFC (21875042, 22279019, 22233002, 22103014), STCSM (21DZ1207102, 21DZ1207103), the National Key Research and Development Program of China (Grant 2018YFA0208600) and Innovation Program for Quantum Science and Technology (2021ZD0303305). This work was also supported by the Program for Eastern Scholars at Shanghai Institutions. This work has also benefited from the 1W1B Beamline at Being Synchrotron Radiation Facility. The authors thank Prof. Lirong Zheng for the assistance in the XAS measurements. The authors also thank Prof. Pengfei Liu for the guidance for the MEA experiments.

## Author contributions

B.Z. and X.X. supervised the project. W.N. conceived the idea and carried out most of the experiments. Z.C. and X.X. carried out DFT calculations. W.N. and W.G. performed XAS measurements. W.N. analyzed the XAS data. W.M., Y.G., J.C., L.Z. and P.W. contributed to the TEM and STEM characterization. W.N. and J.Y. performed the operando ATR-SEIRAS measurements and analyzed the data. Q.Y. performed the ICP-OES measurement. W.N., Z.C., X.X. and B.Z. co-wrote the manuscript. W.G., Y.L., R.H., L.K. and Y.M. assisted with the data analyses and discussions. C.C. provided suggestions on the experimental operations. All authors discussed the results and assisted during manuscript preparation.

## Competing interests

The authors declare no conflict of interest.
