## [Peer Review File · Nature Communications]

nature portfolio

Peer Review FileReviewer comments, first round

Reviewer #1 (Remarks to the Author):

The manuscript by Zhang and co-authors shows a Pb-doped Cu catalyst (Pb-Cu) with numerous atomic Pb-concentrated grain boundaries for n-propanol production in electrochemical reduction of CO. The originality of this work is the implementation of the idea of "atomic size misfit" in CO conversion reaction. The Faradaic efficiency for n-propanol and long-period activity represents excellent benchmarks in terms of absolute selectivity and stability compared to the literature. Besides, spectroscopic and theoretical studies demonstrate a novelty self-adaptive structure which plays a key role in the high C₃ selectivity and stability of the catalyst. This work provides a critical inspiration towards the formation of longer carbon-chain products in electrocatalysis. Thus, I would like to recommend its publication in the Nature Communications after the following issues being addressed.

1. The authors mention that the Pb-Cu reveals an even higher intrinsic activity toward n-propanol as compared to that on the OD-Cu sample. In this section, the ECSA of the commercial Cu nanoparticle also need to be characterized and the intrinsic activity of Cu shall also be provided and compared.
2. Electrochemical environment plays a role in adsorption and reaction energies. By controlling the CO partial pressure, the authors suggested that the enhancement of CO adsorption contributes to n-propanol formation. However, the effect of local pH may also play a role. As the authors indicated in the manuscript that CO concentration is mainly determined by CO partial pressure in the gas feedstock, this can be done by simply changing the pH of electrolyte.
3. The role of Cu⁺ is widely discussed and proposed as the active sites for CO₂RR/CORR (Angew. Chem. Int. Ed. 10.1002/anie.202102606, Adv. Energy Mater. 2020, 2001987). There exists Cu₂O crystalline in XRD. It will be good to discuss whether Cu⁺ participates in the catalytic process.
4. In DFT calculations, the authors suggest *COCOH as the C₂ intermediate. However, both *OCCOH and *CCH₂ are currently suggested as the possible source of C₂ precursor. This shall be addressed in the DFT section.
5. The authors employed operando EXAFS to prove the flexibility of the Pb-Cu surface, which seems reasonable. However, the amplitude of the Cu-Cu path may be caused by both coordination number and disorder of the local structure. Could authors provide the fitting parameters and relative discussions?

Reviewer #2 (Remarks to the Author):

The research work reported a copper based materials with Pb doping and grain boundaries for n-propanol electrosynthesis from CO₂. N-propanol is a valuable industry feedstock with large market, therefore is a significant product from CO₂ reduction. In addition, being a C₃ product, the formation mechanism of n-propanol shows great fundamental importance. Authors have performed extensive characterization of the synthesized samples to identify the morphology (figure 1), local bonding structure with the change of electrode potential (figure 2), as well as the electrochemical performance including long term test for over 100 hours (figure 3). DFT calculations were also performed to confirm the activity bring by Pb atom (figure 4) with some confusion – it's not clear about the x-axis for Figure 4 c,d,e.

Nevertheless, I am not in favour to support acceptance of the work. The authors mentioned that traditionally the Faradaic efficiency for n-propanol production is less than 40%, and their catalyst shows a 47% ± 3% Faradaic efficiency. I would suggest this is incremental improvement judged by Faradaic efficiency (25% improvement). In addition, the theoretical computation shows $\Delta G_{\text{OCCOHCO}}$ is similar to that of Cu(211) and Cu(100), therefore not being a strong evidence of the high C₃ coupling activity.

Reviewer #3 (Remarks to the Author):

In this paper, Niu et al reported an "atomic size misfit" strategy to design Pb-doped Cu catalysts and the Pb-Cu catalysts delivered a high CO-to-n-propanol FE of ~47%. To electrosynthesis C3 product in CORR is challenging and the high n-propanol FE reported in this work is encouraging. Before recommending publication, the following comments should be addressed.

1. Figure 1c is very similar to Supplementary Figure 5d, but Figure 1c is the HAADF-STEM image of Pb-Cu sample while Supplementary Figure 5d is the HAADF-STEM image of Cu sample, based on the captions of these two figures. The authors might want to double check these two figures.
2. To confirm further the lattice expansion of Pd-Cu compared to Cu, could the authors show the partial magnification of the XRD diffraction peaks located at higher angle (e.g. 2 theta ~74°) of Pd-Cu and Cu catalysts and compare the peak shift?
3. The authors claimed that a slightly expanded lattice in the Pb-Cu nanoparticles is observed through atomic-resolution HAADF-STEM characterization. Considering the lattice difference between Pb-Cu (d200 = 18.4 nm) and Cu (d200 = 18.2 nm) is tiny, could the authors provide the resolution of TEM they have used in the work?
4. In Figure 3h, the authors listed two performance points for this research work, but they are confusing. Could the authors describe these two performance points clearly?
5. The authors should also provide n-propanol partial current density in MEA in the main text for readers' information.
6. The authors claimed that Pb-Cu catalyst retains its structure after a long-period operation. But the morphologies shown in TEM images (Supplementary Figure 3a and 34a) of Pb-Cu sample before and after long-period operation are different. The author might want to explain this difference.
7. In Figure 4a, for readers' better understanding, the authors might want to illustrate what atom each colored ball represents.
8. In Figure 4c-e, for each type of energy (binding energy of *CO, formation free energy of *COCO₂H and *COCO₂HCO), the authors listed several energy values for Pd-Cu and Cu(211), respectively. For example, in Figure 4c, the authors provided eight *CO binding energies for Pb-Cu and eight *CO binding energies for Cu(211). The authors might want to give the detailed information for every energy point and describe the difference between these points.
9. The actual length represented by the scale bars in Supplementary Figure 1a and Figure 2a,b should be provided.
10. The authors should give a definition of normalized FE mentioned in Supplementary Figures 18b and 30b.
11. The potentials in Supplementary Figure 27 should be provided.
12. According to authors' description in the main text, in Supplementary Table 7, the FEn-propanol of this work should be 47%.

Reviewer #4 (Remarks to the Author):

In this study, authors reported a Pb-doped Cu catalyst with numerous atomic Pb-concentrated grain boundaries. The Pb-Cu catalyst exhibits a CO-to-n-propanol FE (FE_{propanol}) of 47 ± 3% and a high half-cell energy conversion efficiency (EE) of 25%. A stable CO-to-n-propanol conversation over 100 h with n-propanol FEs above 30% at a total current of 1 A can be obtained in the membrane electrode assembly (MEA) device. Overall, a promising electrocatalyst was demonstrated in the present study. However, the reviewer has several questions regarding the mechanistic insights. Authors should address the experimental design and reaction mechanism in detail.

1.

The CO-to-n-propanol FE was measured in the flow cell and MEA device. The flow cell is three-electrode cell and MEA cell is the two-electrode cell. However, it looks like that the ATR-SEIRAS

measurements were performed in the "H-type cell", which is the three-electrode cell. Authors should provide the cell configuration of in situ SEIRAS/Raman/XAS measurements. Thus, the electrochemical performance and reaction mechanism were studied in different cell configurations. Also, the electrochemical potential in the H-cell/flow-cell is different from that in the MEA cell. It is known that the cell configuration (H-cell, flow cell and MEA cell) results in various electrochemical conditions. How did authors correlate the reaction condition in the flow cell/MEA cell with that in the H-cell (in situ SEIRAS/Raman/XAS studies)? The mechanistic insights are questionable.

2.

The theoretical calculation results show that the CO did not adsorb on the Pb element. In situ SEIRAS results show the different CO species (CO bridge and CO atop) on the Pd-Cu electrocatalyst. The COatop peak on the Pb-Cu surface splits into two bands. Why do not the CO species adsorb on the Pd element? Please provide the explanation and discuss the peak assignments in detail.

3.

Figure 3 shows the in situ SEIRAS/Raman results of Pb-Cu and Cu electrocatalysts. The CO intermediates are obtained at -0.05 V, which is a small bias. Do these results suggest that the CO reduction reaction can be obtained at such low potential? Please address the correlation between the potential-dependent SEIRAS/Raman results and product distribution in detail. The electrochemical performance and reaction mechanism of Pb-Cu electrocatalyst obtained with different cell configurations could be the problem.

4.

Please provide the backward scan of in situ SEIRAS/Raman results. Are the changes in the in situ SEIRAS/Raman measurements reversible?

Summary of Response to the Reviewers

We thank the reviewers for their constructive comments and suggestions to improve the manuscript. We have addressed them point-by-point as documented below. To summarize:

Experimentally:

1. In light of the reviewer's suggestion, we have performed new experiments to estimate the electroactive surface area of the Cu nanoparticles to assess the intrinsic activity of our catalysts more comprehensively.
2. We have carried out new experiments to evaluate the proton concentration dependency on the n-propanol formation.
3. We have provided new data and fitting parameters of the operando EXAFS to discuss the self-adaptive structure of our Pb-Cu catalysts.
4. To address the reviewer's concern, we have added the cell configurations of the operando SEIRAS/Raman/XAS measurements, and have further assessed the CORR performance in different cells.
5. We have carried out new operando Raman and operando ATR-SEIRS experiments with the negative potential shifting and subsequent backward to prove the reversibility of potential-dependent *CO adsorption.

Computationally:

1. In light of the reviewer's suggestion, we have further studied the coupling of *CO and *CCH₂ on Pb-Cu surfaces, and compared the results with those on Cu(211) and Cu(100) surfaces.
2. We have calculated the CO adsorption on Pb elements on Pb-Cu surface to provide more information for peak assignments of the operando SEIRS.

We have revised 3 Figures (Fig. 2, 3 and 4) in the revised Manuscript, 22 Figures (Supplementary Figures 1-2, 5, 8, 16-20, 23-26, 34, 36-37, 41, 47-51) and 4 Tables in the revised Supplementary Information, and revised 10 paragraphs to adequately address the Reviewers' comments.

A point-by-point response to comments is provided below:

Actions (regular font) in Response to Reviewer Comments (*italics*)

Response to Reviewer #1:

Comment 0:

The manuscript by Zhang and co-authors shows a Pb-doped Cu catalyst (Pb-Cu) with numerous atomic Pb-concentrated grain boundaries for n-propanol production in electrochemical reduction of CO. The originality of this work is the implementation of the idea of “atomic size misfit” in CO conversion reaction. The Faradaic efficiency for n-propanol and long-period activity represents excellent benchmarks in terms of absolute selectivity and stability compared to the literature. Besides, spectroscopic and theoretical studies demonstrate a novelty self-adaptive structure which plays a key role in the high C₃ selectivity and stability of the catalyst. This work provides a critical inspiration towards the formation of longer carbon-chain products in electrocatalysis. Thus, I would like to recommend its publication in the Nature Communications after the following issues being addressed.

Response:

We appreciate the reviewer’s positive comments and recommendations. We have carried out new experiments and revised the manuscript to address the concerns of the reviewer. For convenience, the main revisions are discussed in the following point-by-point answers to the reviewer’s questions and marked with blue in the main text.

Comment 1:

The authors mention that the Pb-Cu reveals an even higher intrinsic activity toward n-propanol as compared to that on the OD-Cu sample. In this section, the ECSA of the commercial Cu nanoparticle also need to be characterized and the intrinsic activity of Cu shall also be provided and compared.

Response:

Thanks for the reviewer’s suggestion. We have carried out the electrochemically active surface area (ECSA) experiment on the commercial Cu nanoparticles (Com-Cu, 20 nm) electrode (**Figure R1, i.e., new Supplementary Figure 25 and Table R1, i.e., new Supplementary Table 4**). As the results, the double-layer capacitance of the Com-Cu electrode is 9.77 mF, which is higher than the Pb-Cu electrode (8.85 mF) but lower than that of the OD-Cu sample (12.94 mF). This is coincided with the previous research (*Nat. Catal.*, 2018, **1**, 748–755) that the OD-Cu can induce a larger surface area than the Cu nanoparticles. Moreover, the ECSAs can be change by the doping of other metals on the Cu-base catalysts (*Nat. Commun.*, 2019, **10**, 5186, *Nat. Commun.*, 2020, **11**, 3685).

Figure R1 (i.e., new Supplementary Figure 25). Electrochemical surface area measurement (ECSA) for different catalysts. Determination of double-layer capacitances over a range of scan rates for different catalysts in 1 M KOH saturated with Ar: Pb-Cu (a, d), Cu (b, e) and Com-Cu (c, f). The average capacitance current j ($j = (j_a - j_c)/2$, where j_a and j_c are anodic and cathodic current densities, respectively) for more than 20 circles at 0.17 V (vs. RHE) against the scan rates, C_{dl} values are given by the slopes.

Table R1 (i.e., new Supplementary Table 4). ECSA results of different samples.

Sample	Electric double-layer capacitance (mF)	Surface roughness factor
Com-Cu	9.77	336.9
Cu	12.94	446.2
Pb-Cu	8.85	305.2

The Pb-Cu sample shows a higher n-propanol partial current density, both with respect to geometrical current density and ECSA-normalized current density (Figure R2, i.e., new Supplementary Figure 26), than that of the Com-Cu and the OD-Cu samples. These results are

consistent with our conclusion that the enhanced CO-to-n-propanol activity on the Pb-Cu catalyst should not be attributed to the differences of particle size and surface area.

Figure R2 (i.e., new Supplementary Figure 26). CORR performance of different cathode electrodes. (a) Electric double-layer capacitances and surface roughness factors of the catalysts, calculated by defining the surface roughness factor for electropolished polycrystalline Cu with an electric double layer capacitance of 29 mF as 1 on different electrodes. **(b)** Partial n-propanol current densities with respect to geometrical areas on different electrodes at various potentials. **(c)** Electrochemically active surface area (ECSA)-normalized partial n-propanol current densities on different electrodes.

In light of the reviewer’s comments, we have provided new Supplementary Figures 25-26 on Page 37 and 38 in revised Supplementary Information and revised the following discussions in the manuscript on Line 205, Page 8:

“The CORR performance is also evaluated on the commercial Cu nanoparticles (Com-Cu) (Supplementary Figure 24). In comparison with Com-Cu, our Cu catalyst showed a much higher selectivity of the C₂₊ products including n-propanol. This is consistent with previous reports that OD-Cu can enhance the C₂₊ formation by increasing the local *CO concentration^{11,12,52-54}.”

We have also revised the discussions in the manuscript on Line 210, Page 8:

“Partial current densities were also normalized to electrochemically active surface areas (ECSAs), which reveals an even higher intrinsic activity toward n-propanol on the Pb-Cu as compared to that on both the Cu and the Com-Cu samples (Figures 3e-3f, Supplementary Figures 25-26 and Supplementary Table 4).”

Comment 2:

Electrochemical environment plays a role in adsorption and reaction energies. By controlling the CO partial pressure, the authors suggested that the enhancement of CO adsorption contributes

to n-propanol formation. However, the effect of local pH may also play a role. As the authors indicated in the manuscript that CO concentration is mainly determined by CO partial pressure in the gas feedstock, this can be done by simply changing the pH of electrolyte.

Response:

We appreciate the reviewer's suggestion. We have carried out CORR tests in various electrolytes with different concentration of the KOH ($c[\text{KOH}]$), that is, different pH values (**Figure R3, i.e., new Supplementary Figure 37**). The increased FEs of C_{2+} products were observed as the increasing of $c[\text{KOH}]$ from 0.5M to 1M (corresponding in the pH shift from 13.7 to 14). It is consistent with the literature reports that a higher pH can suppress the HER and promote the C-C coupling process (*Science* 2018, **360**, 783-787).

Figure R3 (i.e., new Supplementary Figure 37). CORR performance on the Pb-Cu catalysts in electrolytes with different KOH concentration at -0.68 V (vs. RHE with no iR compensation).

However, with a further increase of $[\text{KOH}]$ from 1M to 3M (corresponding to the pH of 14 to 14.7), we found that the FEs of both the n-propanol and ethylene drop, whereas acetate formation increased sharply with the $c[\text{KOH}]$ shift in this range. We attribute the acetate increase to the attack of C_2 intermediates for n-propanol in the concentrated OH^- solution (*Nature*, 2014, **508**, 504-507, *J. Am. Chem. Soc.*, 2018, **140**, 9337-9340). As the results, we suggest the $c[\text{KOH}]$ of 1M (pH = 14) to be preferential for n-propanol production.

In light of the reviewer's comments, we have provided new Supplementary Figure 37 on Page 49 in revised Supplementary Information and revised the following discussions in the manuscript on Line 241, Page 9:

“We also assessed the proton concentration dependency on the n-propanol formation. The highest n-propanol FE was obtained in the 1 M KOH, in the range of 0.5 M to 3 M KOH

(Supplementary Figure 37). Further increasing of the solution pH prioritizes the acetate formation due to the increased tendency for the OH⁻ attack towards the C₂ intermediates^{8,55}.”

Comment 3:

The role of Cu⁺ is widely discussed and proposed as the active sites for CO₂RR/CORR (Angew. Chem. Int. Ed. 10.1002/anie.202102606, Adv. Energy Mater. 2020, 2001987). There exists Cu₂O crystalline in XRD. It will be good to discuss whether Cu⁺ participates in the catalytic process.

Response:

We appreciate the reviewer’s suggestion. The role of Cu⁺ has been widely discussed in CO₂RR/CORR, while the electron-deficient Cu sites were suggested to be benefit to the adsorption of specific intermediates to switch the selectivity of the electrocatalysts. However, the Cu surface is easy to be oxidized in the air condition (*Nat. Commun.*, 2019, **10**, 5186), which creates obstacles to the assessment of the Cu electronic structure during the CORR. Therefore, it is a challenge to evaluate the role of Cu⁺ during the reaction.

In this work, we have carried out the operando X-ray absorption spectroscopy to examine the electronic structure of both Cu sites and Pb sites in the Pb-Cu catalysts. As shown in XANES at Cu K-edge and Pb L₃-edge, both elements remained in the metallic state during CORR (Supplementary Figures 11-12). In line with the corresponding Fourier transform spectra, the bond lengths of about 2.54 and 2.73 Å were attributed to the Cu-Cu and the Cu-Pb paths, respectively, and no trace of oxygen was resolved in the lattices of the Pb-Cu catalysts (Supplementary Figure 13 and Supplementary Table 3). Combining these evidences, we tend to suggest that Cu⁺ played a limited role during the CORR in our systems, whereas the trace of Cu₂O characteristic peaks in XRD spectra should be attributed to the partial oxidation of the samples in the air after CORR (*Nat. Energy*, 2022, **7**, 170-176).

In light of the reviewer’s comments, we have revised the legend of the Supplementary Figure 2 on Page 14 in revised Supplementary Information:

“The existence of Cu₂O characteristic peaks were attributed to the partial oxidation of the samples in the air after CORR.”

Comment 4:

*In DFT calculations, the authors suggest *COCOH as the C₂ intermediate. However, both *OCCOH and *CCH₂ are currently suggested as the possible source of C₂ precursor. This shall be addressed in the DFT section.*

Response:

We appreciate the reviewer's suggestion. As recommended, we have calculated the reaction energy for $*CO-^*CCH_2$ coupling on the Pb-Cu surface, and compared the results to that on Cu(211) and Cu(100) surfaces. Similar to the formation of COCOHCO, the reaction free energies (Figure R4f, i.e., new Fig 4f) on two sites of the Pb-Cu surface are comparable to that on the Cu(211) surface, while a higher reaction energy on one site of the Pb-Cu surface is comparable to that on the Cu(100) surface.

Figure R4 (i.e., new Fig 4. DFT calculations). The structures of (a) the Pb-Cu surface and (b) the Cu(211) surface at a high $*CO$ coverage. (c) The binding energy of $*CO$ on (a) the Pb-Cu surface and (b) the Cu(211) surface at a high $*CO$ coverage. As the Pb-doping induced many different low-coordinated Cu atoms, the $*CO$ binding energies of eight CO in (a) are all different, while those in (b) only show two different types of binding energies corresponding to the flat and stepped sites, respectively. The binding energies for each surface are displayed in the decreased order. Among them, $*CO$ (within red circle) on Cu with lowest coordination number shows the strongest binding energies while that on Cu with the highest coordination number (within blue circle) shows the

weakest binding energy. The formation free energy of (d) *COCO₂H, (e) *COCO₂HCO and (f) *COCCH₂ on the Pb-Cu surface and the Cu(211) surface at a high *CO coverage, and the Cu(100) surface at a low *CO coverage. This is consistent with the results of ATR-SEIRAS. The C-C coupling on three different sites of the Pb-Cu surface were calculated and displayed (black points) in activity decreased order in (d), (e) and (f), and compared to those on stepped site of the Cu(211) surface (orange points) and flat site of the Cu(100) surface (yellow points). The most active examples for (g) *COCO₂H, (h) *COCO₂HCO and (i) *COCCH₂ formation on the Pb-Cu surface. All the others can be found in the Supplementary Figure 45-47. The orange, purple, grey, red and white balls present Cu, Pb, C, O and H respectively. The golden balls present the highlighted Cu atoms. The low-coordinated (≤ 7) Cu atoms are highlighted in (a) and (b). The Cu atoms where the C-C coupling happens are highlighted in (g), (h) and (i).

In light of the reviewer's comments, we have added new Supplementary Figure 47, new Figure 4f and 4i and revised the following discussions on Line 321, Page 12 in the manuscript:

“In addition, we also studied the coupling reaction of *CO and *CCH₂, which was also suggested as a possible source of the C₂ precursor. Similar to COCO₂HCO, the reaction free energies (Figure 4f) on two sites of the Pb-Cu surface are comparable to that on the Cu(211) surface, while a higher reaction energy on one site of the Pb-Cu surface is comparable to that on the Cu(100) surface.”

Comment 5:

The authors employed operando EXAFS to prove the flexibility of the Pb-Cu surface, which seems reasonable. However, the amplitude of the Cu-Cu path may be caused by both coordination number and disorder of the local structure. Could authors provide the fitting parameters and relative discussions?

Response:

We appreciate the reviewer's suggestion. To further prove structural differences of the Pb-Cu and the Cu samples, we have supplemented the backward potential scanning of the Cu catalysts for better comparison (Figure R5, i.e., new Supplementary Figure 48). As shown in Figure R5, the amplitude of the Cu-Cu path showed a reversible change on the Pb-Cu samples, while no obvious amplitude variation was detected on the Cu samples with the potential shifting negatively and subsequently backward. We then provided here the fitting parameters of the operando EXAFS results under different potentials with both the Pb-Cu and the Cu catalysts (Figure R6, i.e., new Supplementary Figure 49; Table R2, i.e., new Supplementary Table 7). The EXAFS fitting analysis revealed that the coordination numbers of the Cu-Cu path increased from 9.7 to 11.2 along with the potential shifted from -0.48 V to -0.88 V, and back to 9.8 as the potential returned to -0.48 V. Meanwhile, the Cu-Pb path coordination numbers shifted from 1.2 to 0.7 and back to 1.2 along with the negatively potential shifted and subsequently backward. These results have further proved

the variable coordination structure of the Pb-Cu surface. Moreover, the fitting results have also confirmed the stable structure of the Cu catalysts, where the coordination number remains at 12 under different potentials (Table R3, i.e., new Supplementary Table 8).

Figure R5 (i.e., new Supplementary Figure 48). Operando Cu K-edge EXAFS analyses at different potentials. Fourier-transformed $k^2\chi(k)$ of the Pb-Cu (a) and the Cu (b) catalysts with potential shift from -0.48 V to -0.88 V and backward to -0.48 V (vs. RHE).

Figure R6 (i.e., new Supplementary Figure 49). Operando Cu K-edge EXAFS fitting analyses of the Pb-Cu catalysts. (a) Fourier-transformed $k^2\chi(k)$ of the Pb-Cu catalysts with the potential shift during CORR. (b) The coordination numbers of the Cu-Cu and the Cu-Pb paths under different potentials.

Table R2 (i.e., new Supplementary Table 7). Fitting results of Cu K-edge EXAFS data of the Pb-Cu catalysts under different potentials (V vs. RHE).

Potential (V)	EXAFS	bond	CN	CN _{total}	R (Å)	σ^2 (10^{-3} Å ²)
-0.48	Cu K-edge	Cu-Cu	9.7	10.9	2.54	7.3
		Cu-Pb	1.2		2.72	
-0.68		Cu-Cu	10.6	11.5	2.54	6.2

	Cu-Pb	0.9		2.73	
-0.88	Cu-Cu	11.2	11.9	2.54	6.2
	Cu-Pb	0.7		2.72	
-0.48	Cu-Cu	9.8	11.0	2.54	7.8
	Cu-Pb	1.2		2.72	

Table R3 (i.e., new Supplementary Table 8). Fitting results of Cu K-edge EXAFS data of the Cu catalysts under different potentials (V vs. RHE).

Potential	EXAFS	bond	CN	R (Å)	σ^2 (10^{-3} Å ²)
-0.48		Cu-Cu	12	2.54	9.8
-0.68	Cu K-edge	Cu-Cu	12	2.54	10.2
-0.88		Cu-Cu	12	2.54	9.9
-0.48		Cu-Cu	12	2.54	9.8

In light of the reviewer’s comments, we have provided new Supplementary Figures 48-49 and new Supplementary Tables 7-8 in revised Supplementary Information and revised the following discussions on Line 332, Page 12 in the manuscript:

“This structural flexibility echoes the results from the operando EXAFS and the corresponding fitting analysis under different potentials using the Pb-Cu and the Cu catalysts³⁶ (Supplementary Figures 48-49 and Supplementary Tables 7-8). Hence, the coordination numbers of the Cu-Cu path and the Cu-Pb path exhibit reversible changes with the potential shift on the Pb-Cu sample, revealing a variable coordination structure of the Pb-Cu surface. On the contrary, the Cu catalyst shows a stable structure, whose coordination number remains at 12 under different potentials during CORR.”

Response to Reviewer #2:

Comment 0:

The research work reported a copper based materials with Pb doping and grain boundaries for n-propanol electrosynthesis from CO₂. N-propanol is a valuable industry feedstock with large market, therefore is a significant product from CO₂ reduction. In addition, being a C₃ product, the formation mechanism of n-propanol shows great fundamental importance. Authors have performed extensive characterization of the synthesized samples to identify the morphology (figure 1), local bonding structure with the change of electrode potential (figure 2), as well as the electrochemical

performance including long term test for over 100 hours (figure 3). DFT calculations were also performed to confirm the activity bring by Pb atom (figure 4) with some confusion – it's not clear about the x-axis for Figure 4 c, d, e.

Response:

We thank the reviewer for giving a very nice summary of some the key advances achieved in the present work, such as (1) the significance of producing n-propanol, a valuable industry feedstock with large market, from CO₂ reduction, (2) the great fundamental importance for exploring the formation mechanism of n-propanol, (3) extensive characterization of the synthesized samples to identify the morphology (figure 1), local bonding structure with the change of electrode potential (figure 2), and (4) the electrochemical performance including long term test for over 100 hours (figure 3), (5) as well as DFT calculations to confirm the activity brought by Pb atom (figure 4). We are sorry for the confusion on the legend of Figure 4, which has now been revised for better understanding.

Comment 1:

Nevertheless, I am not in favour to support acceptance of the work. The authors mentioned that traditionally the Faradaic efficiency for n-propanol production is less than 40%, and their catalyst shows a 47% +- 3% Faradaic efficiency. I would suggest this is incremental improvement judged by Faradaic efficiency (25% improvement).

Response:

We agree with the reviewer that our catalyst shows a 47% n-propanol Faradaic efficiency (FE) in a flow-cell reactor. This is superior to the best result reported previously (33% in flow-cell, **Table R4**), such that a 42% improvement has been obtained judged by FE. Moreover, we achieved a half-cell energy efficiency (EE) of 25% (with no iR compensation). With a 70% iR compensation (as reported in *Nat. Commun.*, 2019, **10**, 5186), the half-cell EE reaches 28%, a 1.3× improvement relative to state-of-the-art CO-to-n-propanol electroreduction reports.

As may have been recognized by the reviewer, our work successfully improved the stability of the catalysts, in which we achieved a high stability over 10 hours in a flow-cell reactor, showing a much better stability than the OD-Cu catalyst (**Supplementary Figure 38**) and other OD-Cu based catalysts reported previously (*Nat. Catal.*, 2018, **1**, 748-755, *Nat. Commun.*, 2018, **9**, 4614, *Adv. Mater.*, 2021, **33**, 2103150). Besides, there is no other long-period stable CO-to-n-propanol electroconversion reported in three-electrode systems ever before. Moreover, with a membrane electrode assembly (MEA) device, a 110-h-stable-electrolysis was also obtained.

Overall, our catalyst shows a record CO-to-n-propanol FE of $47 \pm 3\%$ (1.4× improvement versus the previous reports in the three-electrode system), a record half-cell EE of 28% (1.3× improvement), the highest n-propanol current density of 38 mA cm^{-1} , and the best stability (10 h in flow cell, 110 h in MEA).

Also, as may have been recognized by the reviewer, as for the CO-to-C₃ conversion strategy, we have achieved the following new advances in this work:

- (1) For the catalyst design, we have introduced for the first time a “atomic size misfit” strategy to controllably concentrate big atoms (e.g., Pb) in abundant copper (Cu) grain boundaries (GBs), in order to generate and stabilize the low-coordinated Cu sites. In this way, we provided a rational strategy for catalyst design with numerous and stable low-coordinated sites, as a convenient and universal approach to synthesizing Cu-M catalysts, which can be used in various electrochemical reactions. This improves the rational defect-construction of the electrocatalysts with high activity and stability.
- (2) From the scientific scope, we revealed that the high surface *CO coverage plays a key role in the formation of longer carbon-chain products (C_n, n ≥ 3) and found that introducing larger atoms to induce abundant and stable low-coordinated sites is a powerful doping strategy. The findings provide new viewpoints for enhancing carbon-chain growth. Moreover, this new strategy can overcome the drawbacks of some other strategies (e.g., chemical and thermal reduction), in which the low-coordinated sites are unstable during the electrochemical reactions.

To sum up, we achieved an excellent CO-to-n-propanol performance on the Pb-Cu catalysts with a record FE of 47% (1.4× improvement), the highest partial current density in three-electrode systems (38 mA cm⁻¹), the highest cathodic EE of 28% (1.3× improvement) and an extraordinary stability over 100 hours. Additionally, our work on rational construction of the low-coordinated Cu sites is an efficient strategy to develop high-performance catalysts to facilitate the activity and stability of the CO-to-C₃ electroconversion, and make an effective inspiration towards the CORR/CO₂RR to form the longer carbon-chain products. We strongly believe that our work will be of interest to the broad readership of *Nature Communications*.

In light of the reviewer’s comments, we have provided new Supplementary Table 9 on Page 72 in revised Supplementary Information for more comprehensive comparison with previous reports:

Table R4 (i.e., new Supplementary Table 9). Summary of state-of-the-art CORR systems towards n-propanol.

Catalysts	FE (%)	j (mA cm ⁻²)	Half-cell EE (%)*	Full-cell EE (%)	Stability	Remakes	References
Pb-Cu	47	38	28	N/A	10 h	flow-cell	This work

	38	76	N/A	18	110 h	MEA	This work
Ru-Ag-Cu	36	110	N/A	14	102 h	MEA	1
Ag-doped-Cu	33	4.5	21	N/A	N/A	flow-cell	2
Fragmented Cu	20	8.5	12	N/A	1.2 h	flow-cell	3
Nanocavity Cu	21	7.8	12	N/A	N/A	flow-cell	4
OD-Cu	14	20	N/A	N/A	N/A	flow-cell	5
BCF-Cu ₂ O	19	0.9	N/A	N/A	N/A	flow-cell	6
Cu adparticles	23	11	14	N/A	N/A	flow-cell	7

*The half-cell EEs are calculated by 70% iR compensation as the reference 2 for comparison.

References: 1. *Nat. Energy* **7**, 170-176 (2022). 2. *Nat. Commun.* **10**, 5186 (2019). 3. *Nat. Catal.* **2**, 251-258 (2019). 4. *Nat. Catal.* **1**, 946-951 (2018). 5. *Nat. Catal.* **1**, 748-755 (2018). 6. *J. Am. Chem. Soc.* **144**, 12410-12420 (2022). 7. *Nat. Commun.* **9**, 4614 (2018).

Comment 2:

In addition, the theoretical computation shows ΔG_{OCCO^} is similar to that of Cu(211) and Cu(100), therefore not being a strong evidence of the high C_3 coupling activity.*

Response:

The reviewer is right to point out that the calculation results showed that the formation free energies of *OCCO^{*}HCO and *OCCCH₂ of the Pb-Cu surface are comparable to those on Cu(211) and Cu(100) surfaces. As the C_3 coupling activity is around 2 times higher on the Pb-Cu surface than those on Cu(211) and Cu(100) surfaces, conceivably it is the CO coverage θ_{CO} rather than the energy difference that plays a more important role here. Indeed, both experimental (**Figure 2**) and theoretical results (**Figure 4c**) demonstrate that the Pb-doped surface binds CO more strongly than OD-Cu (modeled by Cu(211) in theory), and can hold higher θ_{CO} even under operando condition. Thus, as compared to the OD-Cu, the 2 times higher n-propanol activity of the Pb-Cu catalyst (**Fig. 3f**) has to be attributed to the higher θ_{CO} , whereas an obvious energy differences should have resulted in reactivity differences in orders of magnitude. In addition, this conclusion is also supported by the experimental results that the n-propanol selectivity on the Pb-Cu catalysts highly depends on the CO concentration (**Supplementary Figure 36**).

To avoid confusion, we have added the following sentences in the legend.

“As the Pb-doping induced many different low-coordinated Cu atoms, the *CO binding energies of eight CO in (a) are all different, while those in (b) only show two different types of binding energies corresponding to the flat and stepped sites, respectively. The binding energies for each surface are displayed in the decreased order. Among them, *CO (within red circle) on Cu with lowest coordination number shows the strongest binding energies while that on Cu with the highest

coordination number (within blue circle) shows the weakest binding energy. The formation free energy of (d) *COCO₂H, (e) *COCO₂HCO and (f) *COCCH₂ on the Pb-Cu surface and the Cu(211) surface at a high *CO coverage, and the Cu(100) surface at a low *CO coverage.”

“The C-C coupling on three different sites of the Pb-Cu surface were calculated and displayed (black points) in activity decreased order in (d), (e) and (f), and compared to those on stepped site of the Cu(211) surface (orange points) and flat site of the Cu(100) surface (yellow points). The most active examples for (g) *COCO₂H, (h) *COCO₂HCO and (i) *COCCH₂ formation on the Pb-Cu surface. All the others can be found in the Supplementary Figure 45-47.”

We have also added following sentences on Line 345, Page 12 in the manuscript to further outstand the role of high CO coverage in promoting n-propanol selectivity.

“As compared to OD-Cu, the ~2 times higher n-propanol selectivity of Pb-Cu catalyst (Supplementary Figure 22) is more likely due to the higher θ_{CO} , whereas an obvious energy differences would result in reactivity differences in orders of magnitude. This conclusion is also supported by the experimental results that the n-propanol selectivity on the Pb-Cu catalysts highly depends on the CO concentration (Supplementary Figure 36).”

Response to Reviewer #3:

Comment 0:

In this paper, Niu et al reported an “atomic size misfit” strategy to design Pb-doped Cu catalysts and the Pb-Cu catalysts delivered a high CO-to-n-propanol FE of ~47%. To electrosynthesis C₃ product in CORR is challenging and the high n-propanol FE reported in this work is encouraging. Before recommending publication, the following comments should be addressed.

Response:

We appreciate your positive comments and recommendations. We have carried out new experiments and revised the manuscript to address your concerns below. For convenience, the main revisions are discussed in the following point-by-point answers to the reviewer’s questions and marked with blue in the main text.

Comment 1:

Figure 1c is very similar to Supplementary Figure 5d, but Figure 1c is the HAADF-STEM image of Pb-Cu sample while Supplementary Figure 5d is the HAADF-STEM image of Cu sample, based on the captions of these two figures. The authors might want to double check these two figures.

Response:

Thanks for your helpful reminders and suggestions. We double checked our HAADF-STEM images, and we found that we miswrote the legend of Figures 1b and 1c. Figure 1b and 1c show the topography of the Pb-Cu and the Cu catalysts, respectively.

In light of the reviewer's comments, we have revised the legend of Figures 1b and 1c on Page 18 in the manuscript:

“(b, c) High-angle annular dark-field scanning transmission electron microscopy (HAADF-STEM) images taken from the edge of nanoparticles of the Pb-Cu and the Cu samples, respectively.”

Comment 2:

To confirm further the lattice expansion of Pb-Cu compared to Cu, could the authors show the partial magnification of the XRD diffraction peaks located at higher angle (e.g. $2\theta \sim 74^\circ$) of Pb-Cu and Cu catalysts and compare the peak shift?

Response:

Thanks for your suggestions. We fitted the XRD patterns for both the Pb-Cu and the Cu samples and make a partial magnification of the diffraction peaks located at 43.3° , 50.4° and 74.1° (corresponding to the (111), (200) and (220) planes of Cu), respectively (**Figure R7, i.e., new Supplementary Figure 2**). As compared with the Cu samples, a slight shift ($\sim 0.1^\circ$ for 2θ) for each characteristic peak was observed on the Pb-Cu samples, revealing a slightly larger lattice ($< 0.2\%$ expansion) of the Pb-Cu. However, the XRD data only reflect the bulk structure information of the material. As shown in **Supplementary Figure 7**, the Pb doping mainly exist near the surface of the Pb-Cu catalyst within ~ 30 nm. Therefore, we suppose that the lattice expansion value may increase in the near surface of the Pb-Cu samples. However, it is difficult to be conclusively proved, due to the limited resolution of experimental equipment, e.g., the TEM.

Figure R7 (i.e., new Supplementary Figure 2). Structure analysis of the Pb-Cu and the Cu catalysts. (a, b) SEM patterns of the Pb-Cu (a) and the Cu (b) catalysts, scale bar 200 nm. (c) XRD patterns of the Pb-Cu and the Cu catalysts. (d) The partial magnification and profile fitting of the corresponding diffraction peaks of the (111), (200) and (220) facets in XRD patterns. The existence of Cu_2O characteristic peaks were attributed to the partial oxidation of the samples in the air after CORR.

In light of the reviewer’s comments, we have revised Supplementary Figure 2 on Page 14 in revised Supplementary Information and revised the following discussions on Line 101, Page 4 in the manuscript:

“The XRD diffraction peaks located at $\sim 43.3^\circ$, $\sim 50.4^\circ$ and $\sim 74.1^\circ$ (corresponding to the (111), (200) and (220) planes of Cu) slightly shifts to the lower angle region, revealing a slightly lattice expansion of the Pb-Cu sample.”

Comment 3:

The authors claimed that a slightly expanded lattice in the Pb-Cu nanoparticles is observed through atomic-resolution HAADF-STEM characterization. Considering the lattice difference

between Pb-Cu ($d_{200} = 18.4$ nm) and Cu ($d_{200} = 18.2$ nm) is tiny, could the authors provide the resolution of TEM they have used in the work?

Response:

Thanks for your suggestions. To precisely evaluate the lattice expansion, we have averaged the corresponding spacing for over 20 lattices. We find that the planar space of the Cu(200) in the Pb-Cu and the Cu catalysts are calculated to be 0.184 nm and 0.182 nm, respectively. Thus, we suggested that there is a ~1% lattice expansion in near surface of the Pb-Cu species compared with the Cu species. However, the resolution of the spherical aberration corrected transmission electron microscope (AC-TEM) using in this work is about 0.082 nm. Therefore, the interplanar spacing difference between two samples may still be less than the resolution of the TEM, even after multiplying by 20. So it is indeed hard to rule out the possibility of experimental errors. With this in mind, we decide to revise the narrative in our manuscript, and thanks again for your helpful reminder.

In light of the reviewer's comments, we have revised Supplementary Figure 5 on Page 17 in revised Supplementary Information and revised the following discussions on Line 112, Page 4 in the manuscript:

“High-angle annular dark-field scanning transmission electron microscope (HAADF-STEM) images provide further insights into the microstructures of the Pb-Cu and the Cu catalysts (Figures 1b-1c, Supplementary Figure 5). The Pb-Cu catalyst possesses an average grain size of ~20 nm and abundant GBs, which clearly contrasts with the Cu catalyst.”

Figure R8 (i.e., new Supplementary Figure 5). HAADF-STEM images of the Pb-Cu and the Cu samples. (a, d) High-angle annular dark-field scanning transmission electron microscopy (HAADF-STEM) images taken from the edge of nanoparticles of the Pb-Cu (a) and the Cu (d) samples. Scale bar, 20 nm. **(b, c, e, f)** Atomic-resolution HAADF-STEM images taken from the boxes in (a, d). Scale bar, 5 nm (b, e) and 2nm (c, f).

Comment 4:

In Figure 3h, the authors listed two performance points for this research work, but they are confusing. Could the authors describe these two performance points clearly?

Response:

Thanks for your suggestions. There are two points in Figure R9 (i.e., new Figure 3h) for this work. Point 1 corresponds to the catalytic performance at the highest n-propanol partial current density (Point 1: 34% FE and 47 mA cm⁻² at -0.98 V), while Point 2 corresponds to the performance at the highest n-propanol Faraday efficiency (Point 2: 47% FE and 38 mA cm⁻² at -0.68 V) in a flow-cell reactor.

In light of the reviewer's comments, we have revised Figure 3h and revised the legend of the Figure 3h on Page 20 in the main text:

Figure R9 (i.e., new Figure 3h). Comparison of FEs and partial current densities for n-Propanol between this work and the state-of-the-art CORR catalysts in flow-cell systems. (The Pb-Cu performances in this work: Point 1, at the highest n-propanol Faraday efficiency, -0.68 V vs. RHE; Point 2, at the highest n-propanol current density, -0.98 V vs. RHE).

Comment 5:

The authors should also provide n-propanol partial current density in MEA in the main text for readers' information.

Response:

Thanks for your suggestions. The obtained n-propanol partial current density in MEA was 76 mA cm⁻² with an n-propanol partial current of 378 mA.

In light of the reviewer's comments, we have revised the following discussions on the Line 251, Page 9 in the manuscript:

“At total current of 1 A, the highest n-propanol FE and the full-cell EE of 38% and 18% were obtained, respectively. The n-propanol partial current reaches 378 mA with a current density of 76 mA cm⁻².”

Comment 6:

The authors claimed that Pb-Cu catalyst retains its structure after a long-period operation. But the morphologies shown in TEM images (Supplementary Figure 3a and 34a) of Pb-Cu sample before and after long-period operation are different. The author might want to explain this difference.

Response:

Thanks for your helpful reminder. The scale bar of the **Supplementary Figure 3a** is 50 nm, and it is 500 nm in Supplementary Figure 34a (**Supplementary Figure 41a** in the revised Supplementary Information), so they may seem different. We're sorry to say that the image of 50

nm scale bar we provided (**Supplementary Figure 34b**) was taken in a thick stacking region, making it hard to clearly identify the morphology. Therefore, to avoid confusion, we change the Supplementary Figure 34b taken from the same sample with a clearer morphology (**Figure R10b**, i.e., **Supplementary Figure 41b**), which shows similar morphologies as **Supplementary Figure 3a**.

In light of the reviewer's comments, we have revised Supplementary Figure 41 on Page 53 in revised Supplementary Information:

Figure R10 (i.e., new Supplementary Figure 41). TEM images of the Pb-Cu samples after the 100-h-stability test. (a, b) TEM images. Scale bar, 500 nm (a) and 50 nm (b). (c) HRTEM images of the Pb-Cu particles. Scale bar, 10 nm. Inset, the corresponding Fourier transfer image. (d-f) Atomic-resolution HAADF-STEM images of the Pb-Cu catalyst. Scale bar, 2 nm.

Comment 7:

In Figure 4a, for readers' better understanding, the authors might want to illustrate what atom each colored ball represents.

Response:

Thanks for your suggestions. In Figure 4a and 4b, the orange balls mean Cu atoms; the golden balls represent the low-coordinated Cu atoms and the purple balls refer to Pb atoms.

In light of the reviewer's comments, we have revised the legend of Figure 4a and 4b on Page 21 in the main text as:

“The orange, purple, grey, red and white balls represent Cu, Pb, C, O and H, respectively. The golden balls represent the highlighted Cu atoms. The low-coordinated (≤ 7) Cu atoms are highlighted in (a) and (b). The Cu atoms, where the C-C coupling happens, are highlighted in (g), (h) and (i).”

Comment 8:

*In Figure 4c-e, for each type of energy (binding energy of *CO, formation free energy of *COCO_H and *COCO_HCO), the authors listed several energy values for Pb-Cu and Cu(211), respectively. For example, in Figure 4c, the authors provided eight *CO binding energies for Pb-Cu and eight *CO binding energies for Cu(211). The authors might want to give the detailed information for every energy point and describe the difference between these points*

Response:

Thanks for your suggestions. We have revised the legend of Fig. 4 in the manuscript by adding following description for the energy point:

In light of the reviewer's comments, we have revised the legend of Fig. 4 in the manuscript as:

“The structures of (a) the Pb-Cu surface and (b) the Cu(211) surface at a high *CO coverage. (c) The binding energy of *CO on (a) the Pb-Cu surface and (b) the Cu(211) surface at a high *CO coverage. As the Pb-doping induced many different low-coordinated Cu atoms, the *CO binding energies of eight CO in (a) are all different, while those in (b) only show two different types of binding energies corresponding to the flat and stepped sites, respectively. The binding energies for each surface are displayed in the decreased order. Among them, *CO (within red circle) on Cu with lowest coordination number shows the strongest binding energies while that on Cu with the highest coordination number (within blue circle) shows the weakest binding energy.”

“The C-C coupling on three different sites of the Pb-Cu surface were calculated and displayed (black points) in activity decreased order in (d), (e) and (f), and compared to those on stepped site of the Cu(211) surface (orange points) and flat site of the Cu(100) surface (yellow points). The most active examples for (g) *COCO_H, (h) *COCO_HCO and (i) *COCCH₂ formation on the Pb-Cu surface. All the others can be found in the Supplementary Figure 45-47.”

Comment 9:

The actual length represented by the scale bars in Supplementary Figure 1a and Figure 2a, b should be provided.

Response:

Thanks for your suggestions. We add the scale bars in the legends of **Supplementary Figure 1** and **Figure 2a, b**.

In light of the reviewer's comments, we have revised the Supplementary Figure 1a and Figure 2a, b and the corresponding legend as:

Figure R11 (i.e., Supplementary Figure 1). Structure analysis of CuO nanopowders. (a) SEM and (b) XRD patterns of the as-prepared CuO nanopowders, scale bar 500 nm.

Figure R12 (i.e., Supplementary Figure 2). Structure analysis of the Pb-Cu and the Cu catalysts. (a, b) SEM patterns of the Pb-Cu (a) and the Cu (b) catalysts, scale bar 200 nm. (c) XRD patterns of the Pb-Cu and the Cu catalysts. (d) The partial magnification and profile fitting of the corresponding diffraction peaks of the (111), (200) and (220) facets in XRD patterns. The existence of Cu_2O characteristic peaks were attributed to the partial oxidation of the samples in the air after CORR.

Comment 10:

The authors should give a definition of normalized FE mentioned in Supplementary Figures 18b and 30b.

Response:

Thanks for your suggestions. The normalized FE of each product refers to its selectivity among only carbon-based products by excluding the H_2 contribution (*Nat. Catal.*, 2019, 2, 1124–1131). For example, the normalized FE of n-propanol refers to the selectivity (FE percentage) of n-propanol among all C_{2+} products.

In light of the reviewer’s comments, we have added the reference (*Nat. Catal.*, 2019, 2, 1124–1131) to the legend of Supplementary Figure 23 and 36, and revised the discussion on Line 234, Page 9 as:

“With the decreasing of the surface *CO coverage, the normalized FE¹⁷ (the selectivity among only carbon-base products by excluding the H₂ contribution) of n-propanol on the Pb-Cu decreased from ~48% to ~21% with the ethylene FE increased from ~18% to ~49% (Supplementary Figures 36).”

Comment 11:

The potentials in Supplementary Figure 27 should be provided.

Response:

Thanks for your helpful reminder. The potential values have been added into the new Supplementary Figure 27 (Figure R13, i.e., new Supplementary Figure 33).

Figure R13 (i.e., new Supplementary Figure 34). Comparison of CO-to-n-propanol FEs for various catalysts.

Comment 12:

According to authors’ description in the main text, in Supplementary Table 7, the FE_{n-propanol} of this work should be 47%.

Response:

Thanks for your suggestion.

In light of the reviewer’s comments, we have revised the n-propanol FE of this work to 47% in Supplementary Table 7 (i.e., new Supplementary Table 9).

Table R5 (i.e., new Supplementary Table 9). Summary of state-of-the-art CORR systems towards n-propanol.

Catalysts	FE (%)	j (mA cm ⁻²)	Half-cell EE (%)*	Full-cell EE (%)	Stability	Remakes	References
Pb-Cu	47	38	28	N/A	10 h	flow-cell	This work
	38	76	N/A	18	110 h	MEA	This work
Ru-Ag-Cu	36	110	N/A	14	102 h	MEA	1
Ag-doped-Cu	33	4.5	21	N/A	N/A	flow-cell	2
Fragmented Cu	20	8.5	12	N/A	1.2 h	flow-cell	3
Nanocavity Cu	21	7.8	12	N/A	N/A	flow-cell	4
OD-Cu	14	20	N/A	N/A	N/A	flow-cell	5
BCF-Cu ₂ O	19	0.9	N/A	N/A	N/A	flow-cell	6
Cu adparticles	23	11	14	N/A	N/A	flow-cell	7

*The half-cell EEs are calculated by 70% iR compensation as the reference 2 for comparison.

References: 1. *Nat. Energy* **7**, 170-176 (2022). 2. *Nat. Commun.* **10**, 5186 (2019). 3. *Nat. Catal.* **2**, 251-258 (2019). 4. *Nat. Catal.* **1**, 946-951 (2018). 5. *Nat. Catal.* **1**, 748-755 (2018). 6. *J. Am. Chem. Soc.* **144**, 12410-12420 (2022). 7. *Nat. Commun.* **9**, 4614 (2018).

Response to Reviewer #4:

Comment 0:

In this study, authors reported a Pb-doped Cu catalyst with numerous atomic Pb-concentrated grain boundaries. The Pb-Cu catalyst exhibits a CO-to-n-propanol FE (FE_{propanol}) of 47 ± 3% and a high half-cell energy conversion efficiency (EE) of 25%. A stable CO-to-n-propanol conversation over 100 h with n-propanol FEs above 30% at a total current of 1 A can be obtained in the membrane electrode assembly (MEA) device. Overall, a promising electrocatalyst was demonstrated in the present study. However, the reviewer has several questions regarding the mechanistic insights. Authors should address the experimental design and reaction mechanism in detail.

Response:

Thanks for the reviewer's positive comments. We have carried out new experiments and revised the manuscript to address the reviewer's concerns regarding the mechanistic insights. For convenience, the main revisions are discussed in the following point-by-point answers to the reviewer's questions and marked with blue in the manuscript.

Comment 1:

The CO-to-n-propanol FE was measured in the flow cell and MEA device. The flow cell is three-electrode cell and MEA cell is the two-electrode cell. However, it looks like that the ATR-

SEIRAS measurements were performed in the “H-type cell”, which is the three-electrode cell. Authors should provide the cell configuration of in situ SEIRAS/Raman/XAS measurements. Thus, the electrochemical performance and reaction mechanism were studied in different cell configurations. Also, the electrochemical potential in the H-cell/flow-cell is different from that in the MEA cell. It is known that the cell configuration (H-cell, flow cell and MEA cell) results in various electrochemical conditions. How did authors correlate the reaction condition in the flow cell/MEA cell with that in the H-cell (in situ SEIRAS/Raman/XAS studies)? The mechanistic insights are questionable.

Response:

Thanks for your comments. In this work, we mainly use a flow-cell reactor to evaluate the catalytic performance of the Pb-Cu catalysts, in which the Pb doping exhibits a promising enhancement of n-propanol FE with the peak value of 47%, and a higher stability than the Cu catalysts (10 h vs. less than 3 h) (**Supplementary Figure 38**). A MEA electrolyzer was also used to further evaluate the stability of the Pb-Cu catalysts by excluding the effects of the carbon paper flood (*Nat. Catal.*, 2022, **5**, 251–258, *Joule.*, 2020, **12**, 011). As have noticed by the reviewer, the MEA is a two-electrode system and the electrode potential cannot be obtained. The electrode potential and reaction condition are truly different from the flow-cell, but a high n-propanol FE was also obtained as 38% (**Supplementary Figure 39 and Supplementary Table 9**). Therefore, we suppose that in the MEA electrolyzer, the Pb-Cu catalysts still exhibit a good ability for high *CO coverage to promote the C₃ formation.

In light of the reviewer’s comments, we have revised the discussions on Line 253, Page 9 in the manuscript:

“The catalytic performance in an MEA electrolyzer is not totally in line with that in flow-cell. As a high n-propanol FE was also obtained in an MEA electrolyzer, we supposed that the Pb-Cu catalysts still exhibit a good ability of enhancing surface *CO coverage to promote the C₃ formation as that in flow-cell.”

Meanwhile, we mainly evaluated the mechanistic insights of the Pb-Cu catalysts in the three-electrode system because the highest C₃ selectivity was obtained in a flow-cell reactor.

We have followed the Reviewer’s suggestions and provide the cell configuration of the operando SEIRAS/Raman/XAS measurements, respectively. Moreover, we have evaluated the catalytic performance in different cell configurations for mechanistic research.

(1) Cell configuration for the operando XAS/SEIRAS/Raman measurements

(i) **Operando XAS cell.** The operando XAS measurements were performed with a home-made flow cell reactor, which is similar to the flow-cell we used for preference measurements (e.g., the CO-to-

n-propanol FE). The only difference is that the outer surface of the gas chamber was replaced with the Kapton tape (**Figure R14, i.e., new Supplementary Figure 8**). The reaction condition is totally the same as that in the catalytic performance evaluation.

Figure R14 (i.e., new Supplementary Figure 8). Cell configuration of the flow-cell type reactors. A schematic diagram of the flow-cell type electrolyzer configuration for the catalytic performance (a) and the operando XAS measurements (b). (c) An image of the flow-cell type electrolyzer during operando XAFS test.

(ii) **Operando ATR-SEIRAS cell.** The operando ATR-SEIRAS measurements were performed with a H-type cell (**Figure R15, i.e., new Supplementary Figure 16**), different Cu-based catalysts coated on the Au/Si substrate was used as the working electrode, a Hg/HgO electrode and a graphite rod were applied as the reference and counter electrodes, respectively.

Figure R15 (i.e., new Supplementary Figure 16). Cell configuration of the H-type cell for operando ATR-SEIRAS.

(iii) **Operando Raman cell.** The operando Raman measurements were carried out within a modified flow cell reactor and a water immersion objective. The Raman cell (**Figure R16, i.e., new Supplementary Figure 18**) has a similar reaction condition to our flow-cell reactor with a triple-phase interface that allows the gas reactant to contact the catalyst-electrolyte interface, overcoming the mass transfer limitation of CO during the reaction.

Figure R16 (i.e., new Supplementary Figure 19). Cell configuration of the modified flow-cell type reactors for the operando Raman spectroscopy.

(2) Evaluation of catalytic performance in the ATR-SEIRAS and Raman cell.

As suggested, we have evaluated the CORR performance of the Pb-Cu and the Cu catalysts in both SEIRAS and Raman cells (**Figure R17-R18**). We achieved peak n-propanol FEs as ~37% and 43% at -0.68 V (vs. RHE) in the SEIRAS and Raman cell, respectively. Besides, in the potential range of -0.58V to -0.78V, the Pb-Cu catalysts show enhanced n-propanol selectivity compared with the Cu catalysts. Furthermore, in both cells, the n-propanol and ethylene FEs were promoted and the ethanol was suppressed after the Pb doping, similar to the performance in the flow-cell reactor (**Supplementary Figure 22**). These results suggest that the reaction mechanism of the Pb-Cu catalysts in various cell configuration we used for mechanism insights may be comparable, where the key conclusion remains valid. In other words, the Pb-Cu catalysts can enhance surface *CO coverage to promote the C₃ formation.

Figure R17 (i.e., Supplementary Figure 50). CORR performances of catalysts in the ATR-SEIRAS cell. The FEs of the CORR products on (a) the Pb-Cu and (b) the Cu catalysts under different potentials.

Figure R18 (i.e., Supplementary Figure 51). CORR performances of catalysts in the Raman cell. The FEs of the CORR products on (a) the Pb-Cu and (b) the Cu catalysts under different potentials.

In light of the reviewer's comments, we have provided new Supplementary Figures 8, 16, 19, 50-51 in revised Supplementary Information and revised the following discussions on Line 178, Page 7; Line 224, Page 8 and Line 265, Page 10 in revised Supplementary Information:

“The Pb-Cu GDE (same as the electrochemical measurements) was carried out with a chronoamperometry process at -0.68 V (vs. RHE) in a home-made flow-cell type reactor for the operando XAS measurements, similar to the flow cell used for preference measurements, and the only difference is that the outer surface of the gas chamber was replaced with the Kapton tape (Supplementary Figure 8).”

“To make our mechanistic insights based on ATR-SEIRAS more convincible, the catalytic performance was also evaluated in the ATR-SEIRAS cell. We achieved a peak n-propanol FE of ~37% at -0.68 V (vs. RHE), ~2 times higher n-propanol selectivity than that of the Cu catalysts (17%). In addition, in the potential range of -0.58V to -0.78V, the Pb-Cu catalysts showed much enhanced n-propanol selectivity compared with the Cu catalysts. Furthermore, both of the n-propanol and ethylene FEs were promoted and the ethanol was suppressed after the Pb doping, similar to the performance obtained in the flow-cell reactor (Supplementary Figure 50).”

“The catalytic performance was also evaluated in the Raman cell. In the potential range from -0.58V to -0.78V, the peak n-propanol FE of ~43% was obtained at -0.68 V (vs. RHE) on the Pb-Cu catalysts, ~2 times higher n-propanol than that on the Cu catalysts. Besides, both of the n-propanol and ethylene FEs were promoted and the ethanol was suppressed after the Pb doping, similar to the performance in the flow-cell reactor and ATR-SEIRS cell (Supplementary Figure 51).”

Comment 2:

The theoretical calculation results show that the CO did not adsorb on the Pb element. In situ SEIRAS results show the different CO species (CO bridge and CO atop) on the Pb-Cu electrocatalyst. The COatop peak on the Pb-Cu surface splits into two bands. Why do not the CO species adsorb on the Pb element? Please provide the explanation and discuss the peak assignments in detail.

Response:

Thanks for your suggestions. By means of DFT calculations, it was found that the optimized structure for CO adsorption on Pb element has a Pb-CO distance of 3.5 Å with an adsorption energy close to 0. This suggested that CO does not adsorb on the Pb site, which, thus, was not considered in peak assignments.

Figure R19 (i.e., Supplementary Figure 17). The optimized structure for CO adsorption on the Pb site of the Pb-Cu surface by DFT calculations.

Besides, due to the different *CO coverage on the Pb-Cu catalyst compared to the Cu catalyst, the adsorption configuration of *CO is apparently different. The adsorption bands in the range of 2027-2080 cm^{-1} (*CO_{atop}) split into separate bands in the ATR-SEIRS spectra. The higher frequency band (HFB) at around 2075-2080 cm^{-1} indicates a stronger binding of *CO to the Cu sites. We attribute this band to *CO on the low-coordinated sites, for example, the grain boundary sites observed by us on the Pb-Cu samples. On the contrary, there is only one *CO_{atop} band on the Cu sample, located at around 2027-2067 cm^{-1} , showing a generally weaker *CO binding to the Cu surface as evidenced by the CO-TPD and DFT results. Furthermore, the presence of the HFB also reveals a high *CO surface coverage on the catalyst. This result is in line with the recent reports (*Nat. Nanotechnol.*, 2023, 18, 299-306; *ACS Catal.*, 2020, 10, 12, 6908–6923).

In light of the reviewer's comments, we have provided new Figure R19 as Supplementary Figure 17, and added the following discussions in the manuscript on Line 150, Page 6:

“Note that, CO adsorption on Pb was not considered in peak assignments, since CO was found not to adsorb on the Pb sites by the DFT calculations (Supplementary Figure 17).”

Comment 3:

Figure 3 shows the in situ SEIRAS/Raman results of Pb-Cu and Cu electrocatalysts. The CO intermediates are obtained at -0.05 V, which is a small bias. Do these results suggest that the CO reduction reaction can be obtained at such low potential? Please address the correlation between the potential-dependent SEIRAS/Raman results and product distribution in detail. The electrochemical performance and reaction mechanism of Pb-Cu electrocatalyst obtained with different cell configurations could be the problem.

Response:

Thanks for your comments. We might not have articulated clearly in the main text, which could have confused the reviewer about the spectroscopy results. Since CO is the reactant in our study, once CO molecules are introduced into the solution and reach the electrode surface, they will adsorb on the catalysts, even at a low potential. Under small biases, the CORR cannot be fully set up due to the sluggish kinetics (**Figure R20**). There is almost no *CO consumption, such that the *CO adoption peaks are intense. As the potential shifts more negatively, the kinetics is accelerated, and the *CO peaks are decreased. These results are in line with recent reports. (*Nat. Catal.*, 2022, **5**, 251–258, *J. Am. Chem. Soc.*, 2022, 144, 48, 22202–22211).

Specifically, in the Figure 3a and 3b, the CO adsorption peaks are intense when the potential is lower than -0.5 V. During the negative scanning of the potential, the *CO peak, especially the *CO_{atop} decreased rapidly. Similar results were observed in the Raman spectra, where the *CO peaks decreased as the potential scanned negatively. We attribute this phenomenon to the *CO consumption because of the C-C coupling processes on both the Pb-Cu and the Cu catalysts at large biases.

To further prove our conclusions, we conducted the backward scan on the Pb-Cu and the Cu catalysts in both operando Raman (**Figures R21-R22, i.e., new Figure 2c and new Supplementary Figure 20**) and ATR-SEIRAS (**Figure R23, i.e., new Supplementary Figure 18**). The *CO peaks decreased as the potential scanned negatively, and increased again as the potential shifted back to a small bias. These results convinced us that the detected *CO species are the precursors for the subsequent C-C coupling at reaction potentials.

Figure R20. Cyclic-voltammetry analyses of the Pb-Cu catalyst in the ATR-SEIRAS cell.

Figure R21 (i.e., new Figure 2c). The operando Raman spectra on the Pb-Cu electrode under CORR. The spectra were recorded on the same point at different potentials from -0.28 V to -0.88 V and backward to -0.28V (vs. RHE).

Figure R22 (i.e., new Supplementary Figure 20). The operando Raman spectra on the Cu electrode under CORR. The spectra were recorded on the same point at different potentials from -0.28 V to -0.88 V and backward to -0.28V (vs. RHE).

Figure R23 (i.e., new Supplementary Figure 18). The operando ATR-SEIRAS spectra on different electrodes under CORR. The spectra were recorded on the Pb-Cu (a) and the Cu (b) catalysts with negative staircase potential scan from -0.10 V to -0.80 V and backward to -0.10V (vs. RHE).

In light of the reviewer’s comments, we have provided new Figure 2c in the main text. We have also provided new Supplementary Figures 18, 20 in revised Supplementary Information and revised the following discussions in the manuscript on Line 162, Page 6 and Line 174, Page 7:

“Additionally, it was found that the adsorption bands were strong under small biases, and as the potential shifts more negatively, the *CO peaks decreased rapidly. We further provided the backward scan of the ATR-SEIRAS on both catalysts (Supplementary Figure 18). The *CO peaks decreased as the potential scanned negatively, and increased again as the potential shifted back to a small bias. These results confirmed that the detected *CO species are the precursors for the subsequent C-C coupling processes at reaction potentials in CORR.”

“Consistent with the ATR-SEIRAS results, the *CO_{bridge} species were only obtained on the Pb-Cu catalyst, and the *CO bands changed reversibly as the potential shifted negatively and subsequently backward.”

Comment 4:

Please provide the backward scan of in situ SEIRAS/Raman results. Are the changes in the in situ SEIRAS/Raman measurements reversible?

Response:

Thanks for your helpful suggestions. As the response to the comment 3, the backward scanning of the operando SEIRAS/Raman measurements results are provided in **Figures R21-R23**. As shown

by the results, the *CO peaks changed reversibly. Specifically, the*CO peaks decreased as the potential scanned negatively, and increased again as the potential shifted back to a small bias.

Reviewer comments, further round

Reviewer #1 (Remarks to the Author):

The authors have fully addressed my concerns as well as other reviewers'. The manuscript is now in a good shape.

Reviewer #3 (Remarks to the Author):

The authors addressed my comments. I recommend its publication on Nature Communications.

Reviewer #4 (Remarks to the Author):

Authors have addressed the questions properly. I would like to recommend this study for further publication after the following minor issue being addressed.

The reaction mechanism of electrocatalysts on the GDE was studied in the flow-cell using in situ Raman/XAS. However, the ATR-SEIRAS measurement was performed using H-type cell "without GDE". Although elucidating the CO adsorption behavior is main purpose in the Raman/SEIRAS measurements, the reaction conditions including the mass diffusion are different with and without GDE. Please provide the detailed discussion and address this issue clearly in this study.

Actions (regular font) in Response to Reviewer Comments (*italics*)

Response to Reviewer #1:

The authors have fully addressed my concerns as well as other reviewers'. The manuscript is now in a good shape.

Response:

We appreciate the reviewer's efforts for peer-review.

Response to Reviewer #3:

The authors addressed my comments. I recommend its publication on Nature Communications.

Response:

We appreciate the reviewer's recommendation.

Response to Reviewer #4:

Comment 0:

Authors have addressed the questions properly. I would like to recommend this study for further publication after the following minor issue being addressed.

Response:

We appreciate the reviewer's recommendation. In light of the reviewer's comments, we revised the manuscript and provided more detailed discussion in the main manuscript.

Comment 1:

The reaction mechanism of electrocatalysts on the GDE was studied in the flow-cell using in situ Raman/XAS. However, the ATR-SEIRAS measurement was performed using H-type cell "without GDE". Although elucidating the CO adsorption behavior is main purpose in the Raman/SEIRAS measurements, the reaction conditions including the mass diffusion are different with and without GDE. Please provide the detailed discussion and address this issue clearly in this study.

Response:

We agree with the reviewer's comment that the mass diffusion is quite different with and without a GDE. So that, in Figure 2a and 2b, the peak sharply reduces with the potential shifts negatively, due to the accelerated CO consumption via the CORR. The CO_{atop} peak almost disappears at -0.50 V, indicating severe mass transport limitation. On the contrary, in the Raman spectra, the CO_{atop} peak disappears at -0.78 V, a much larger bias. This might indicate the mass transport difference between the GDE and a smooth Au/Si electrode.

In light of the reviewer's comments, we have provided more detailed discussion in the main manuscript, besides, we have added the reference (*J. Am. Chem. Soc.*, 2022, 144, 48, 22202–22211) to this discussion.

“It is to note that the potential where the *CO_{atop} peaks almost disappear in Raman (-0.78 V, Figure 2c) is more negative than that in ATR-SEIRAS spectra (-0.5 V, Figures 2a-2b). This might be due to the mass transport difference between the gas diffusion electrode and a smooth Au/Si electrode⁴⁰.”